# TEAD1 is crucial for developmental myelination, Remak bundles, and functional regeneration of peripheral nerves

**Matthew Grove[1], Hyukmin Kim[1], Shuhuan Pang[1], Jose Paz Amaya[2], Guoqing Hu[3], Jiliang Zhou[3], Michel Lemay[2], Young-Jin Son[1]\***

[1]Department of Neural Sciences, Shriners Hospitals Pediatric Research Center, Lewis Katz School of Medicine, Temple University, Philadelphia, United States; [2]Department of Bioengineering, Temple University, Philadelphia, United States; [3]Department of Pharmacology & Toxicology, Medical College of Georgia, Augusta University, Augusta, United States

**Abstract** Previously we showed that the hippo pathway transcriptional effectors, YAP and TAZ, are essential for *Schwann cells* (SCs) to develop, maintain and regenerate myelin . Although TEAD1 has been implicated as a partner transcription factor, the mechanisms by which it mediates YAP/TAZ regulation of SC myelination are unclear. Here, using conditional and inducible knockout mice, we show that TEAD1 is crucial for SCs to develop and regenerate myelin. It promotes myelination by both positively and negatively regulating SC proliferation, enabling Krox20/Egr2 to upregulate myelin proteins, and upregulating the cholesterol biosynthetic enzymes FDPS and IDI1. We also show stage-dependent redundancy of TEAD1 and that non-myelinating SCs have a unique requirement for TEAD1 to enwrap nociceptive axons in Remak bundles. Our findings establish TEAD1 as a major partner of YAP/TAZ in developmental myelination and functional nerve regeneration and as a novel transcription factor regulating Remak bundle integrity.

**\*For correspondence:**
yson@temple.edu

**Competing interest:** The authors declare that no competing interests exist.

## Editor's evaluation

This important study demonstrates that the transcription factor TEAD1 is required for the function of Yap/Taz in Schwann cells, with conditional mouse mutants having very similar dysmyelinated phenotypes. Convincing histological evidence is shown for the role of TEAD1 itself, leaving open the function of other TEAD proteins in this system. This study will nevertheless be of great interest to researchers in the field of peripheral nerve development.

## Introduction

In the mammalian peripheral nervous system, all motor and sensory axons are associated with Schwann cells (SCs). Axons larger than 1 µm diameter are enveloped by myelinating SCs that form multilamellar myelin sheaths with a thickness proportional to axon caliber (*Jessen and Mirsky, 2005*; *Salzer, 2015*). Axons smaller than 1 µm diameter, the unmyelinated nociceptors (i.e., C-fibers), are organized in clusters called Remak bundles, in which each C-fiber is ensheathed by a cytoplasmic pocket formed by non-myelinating SCs, also known as Remak SCs (*Beirowski et al., 2014*; *Harty and Monk, 2017*). Both myelinating and non-myelinating SCs provide metabolic and functional support essential for structural and functional integrity of peripheral nerves. Failure to initiate or maintain proper axon-SC

association may lead to life-threatening paralysis and peripheral neuropathy (*Scherer and Wrabetz, 2008*). Myelinating SCs form and maintain myelin via a set of key transcription factors (TFs) and regulators (*Bremer et al., 2011*; *Decker et al., 2006*; *Finzsch et al., 2010*; *He et al., 2010*; *Jaegle et al., 2003*; *Kao et al., 2009*; *Le et al., 2005a*). However, the transcriptional regulation of myelinating SCs remains incompletely understood, and that of non-myelinating SCs is largely unknown (*Harty and Monk, 2017*).

Recent mouse genetic studies surprisingly revealed that YAP (Yes-associated protein, also known as YAP1) and its paralogue, TAZ (Transcriptional coactivator with PDZ-binding motif, also known as WWTR1), which are potent oncoproteins and transcriptional regulators in the Hippo signaling pathway, play crucial roles in developing, maintaining, and regenerating myelination of peripheral nerves. Both YAP and TAZ (YAP/TAZ) are expressed initially in all immature SCs and then selectively in myelinating SCs (*Grove et al., 2017*). The processes for which they are required include: timely sorting of large axons and associated SC proliferation (*Grove et al., 2017*; *Poitelon et al., 2016*); initiation of SC differentiation or myelination (*Deng et al., 2017*; *Grove et al., 2017*); myelin maintenance in adults *Grove et al., 2017*; and remyelination following nerve injury (*Grove et al., 2020*; *Jeanette et al., 2021*). YAP/TAZ initiate and maintain myelination in part by upregulating Krox20 (Egr2; *Grove et al., 2017*; *Lopez-Anido et al., 2016*), which is widely recognized as the master TF promoting peripheral myelination by SCs. Krox20 is required to develop and maintain myelin (*Decker et al., 2006*; *Topilko et al., 1994*), and likely to promote myelin sheath growth (*Le et al., 2005a*). It acts through direct transcriptional upregulation of myelin proteins and membrane lipids (*Jang et al., 2010*; *Jang and Svaren, 2009*; *LeBlanc et al., 2006*).

YAP/TAZ are transcriptional co-activators that contain a transcriptional activation domain, but no DNA binding domain. They regulate gene expression by binding to partner TFs, typically members of the TEA domain family (TEAD1-4; *Wu et al., 2008*; *Zhao et al., 2008*). Conversely, TEADs on their own are unable to activate or repress transcription and require the help of co-factors such as YAP/TAZ to act (*Li et al., 2010*; *Tian et al., 2010*). Partner TFs such as TEADs are required to maintain YAP/TAZ in the nucleus, which keeps them transcriptionally active (*Diepenbruck et al., 2014*; *Kim et al., 2020*; *Kofler et al., 2018*; *Yao et al., 2018*). Recent single-cell transcriptional analysis indicates that SCs express all four TEADs (TEAD1/2/3/4) and that their expression peaks at different times during development (*Gerber et al., 2021*). Moreover, TEAD1-4 show unique, redundant, and/or opposing roles in regulating proliferation and differentiation of various cell types; these roles can also change as a function of developmental stage (*He et al., 2018*; *Mukhtar et al., 2020*). It remains to be elucidated how and which TEAD(s) in SCs mediate transcriptional regulation of YAP/TAZ for the development, maintenance, and/or regeneration of myelination after nerve injury.

In some cell types, TEAD1 serves as the main partner TEAD and plays non-redundant roles (*Liu et al., 2019*; *Wen et al., 2019*). In accordance, in SCs, TEAD1 associates with YAP/TAZ for upregulating myelin proteins as indicated by ChiP, co-IP, and transcription assays (*Fernando et al., 2016*; *Grove et al., 2017*; *Lopez-Anido et al., 2016*; *Lopez-Anido et al., 2015*). Notably, however, unlike YAP/TAZ, which are barely detectable in non-myelinating SCs, TEAD1 is clearly expressed in all SC nuclei (*Grove et al., 2017*), suggesting that, independently of YAP/TAZ, TEAD1 plays an unknown role in non-myelinating SCs. In this study, we generated conditional and inducible knockout mice lacking TEAD1 in SCs to investigate the roles of TEAD1 in myelinating and non-myelinating SCs. We found that TEAD1 is required for myelinating SCs to form, grow, and regenerate myelin, and for non-myelinating SCs to enwrap C-fibers in Remak bundles during development and homeostasis. We also found that YAP/TAZ-TEAD1 promotes myelination by dually regulating SC proliferation, by enabling induced Krox20 to upregulate myelin proteins, and by upregulating cholesterol biosynthetic enzymes.

## Results

### Multiple TEADs are expressed in *Schwann cells* of WT and *Tead1* cKO mice

To investigate the roles of SC TEAD1 in developing nerves, we conditionally ablated *Tead1* by crossing *Tead1*[fl/fl] mice (*Wen et al., 2017*) with mice carrying *Mpz*-Cre, which is active from E13.5 in SCs (*Feltri et al., 1999*). Immunostaining of P22 WT and *Mpz-Cre;Tead1*[fl/fl] (hereafter *Tead1* cKO) sciatic nerves (SNs) showed that TEAD1 was expressed in the nuclei of 100% of WT SCs and was undetectable

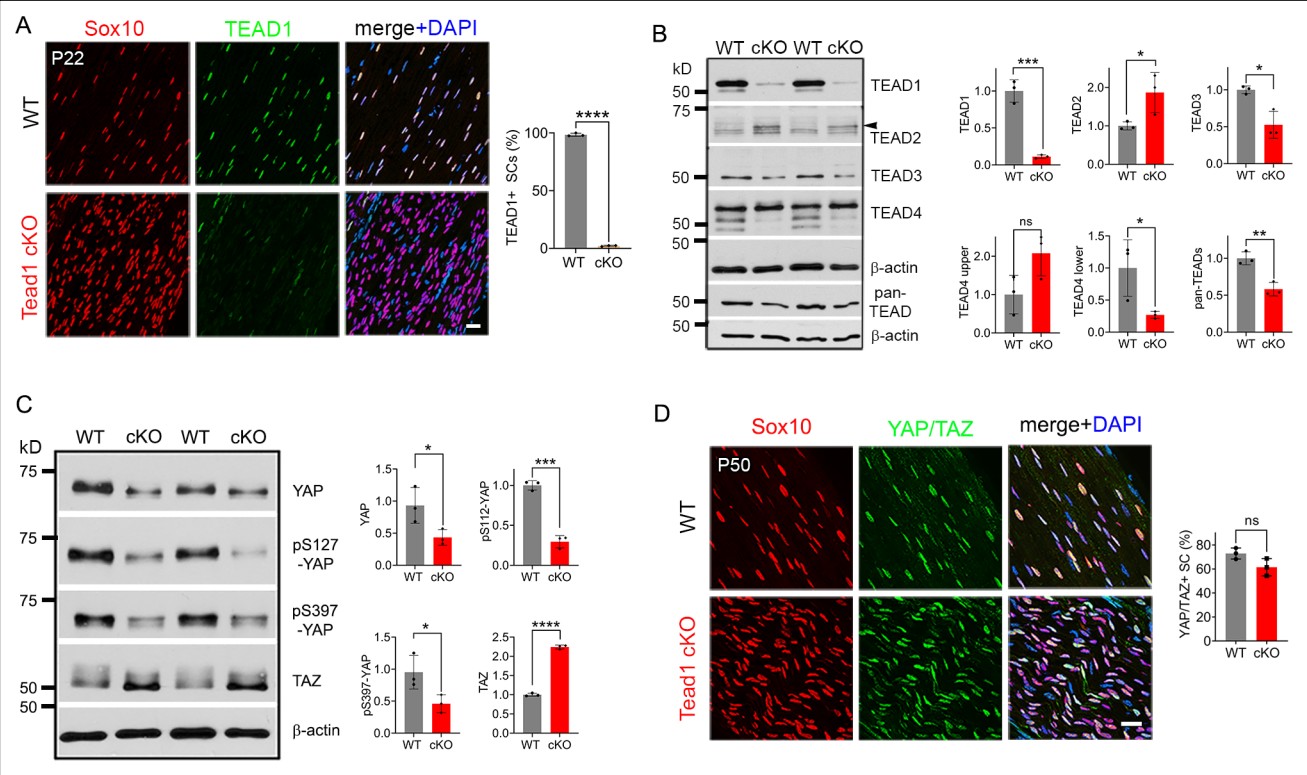

**Figure 1.** Multiple TEADs are expressed in *Schwann cells* of WT and *Tead1* cKO mice. (**A**) Immunostaining and quantification of TEAD1 +SCs on longitudinal sections of P22 WT and *Tead1* cKO SNs. SC nuclei are identified by Sox10. All cell nuclei are marked by DAPI. n=3 mice per genotype, ****$p < 0.0001$, Student's unpaired t-test. Scale bar: 20 µm. (**B**) Western blotting of P50 WT and *Tead1* cKO SN lysates with anti-TEAD1, -TEAD2, -TEAD3, -TEAD4, and -pan TEAD. TEAD expression is normalized to that of *β-actin* as an internal control, and WT expression is arbitrarily given the value 1. n=3 mice per genotype. ns, $p > 0.05$; *$p \leq 0.05$, **$p \leq 0.005$, ***$p \leq 0.001$, Student's unpaired t-test. (**C**) Western blotting of P50 WT and *Tead1* cKO SN lysates with anti-YAP, -phospho-YAPs (Ser127 and Ser397), and -TAZ. YAP/TAZ expression is normalized to that of *β-actin* as an internal control, and WT expression is arbitrarily given the value 1. n=3 mice per genotype. *$p \leq 0.05$, ***$p \leq 0.001$, ****$p < 0.0001$, Student's unpaired t-test. (**D**) Immunostaining of YAP/TAZ +SCs on longitudinal sections of P50 WT and *Tead1* cKO SNs. SC nuclei are identified by Sox10. All cell nuclei are marked by DAPI. n=3 mice per genotype, ns, $p > 0.05$, Student's unpaired t-test. Scale bar: 20 µm.

The online version of this article includes the following source data and figure supplement(s) for figure 1:

**Source data 1.** This zip archive contains source files for graphs in *Figure 1A, B, C and D* and uncropped labeled or unlabeled blots of *Figure 1B and D*.

**Figure supplement 1.** Western blotting of P5, P38, P85 WT, and P38 *Tead1* cKO SN lysates with anti-TEAD2.

**Figure supplement 1—source data 1.** This zip archive contains uncropped labeled or unlabeled blots in *Figure 1—figure supplement 1*.

in >97% *Tead1* cKO SCs (*Figure 1A*). The specificity of the TEAD1 immunostaining and highly efficient deletion were further confirmed by western blotting (*Figure 1B*).

To investigate the protein expression of other TEADs, we used antibodies to TEAD2, 3, and 4, and a pan-TEAD antibody marketed as recognizing all four TEADs (TEAD1/2/3/4). Western blot analysis of SN lysates showed that anti-TEAD3 and anti-pan TEAD both contained a main ~50 kD band that was decreased in *Tead1* cKO (*Figure 1B*). The pan-TEAD antibody may therefore bind preferentially to TEAD3. Anti-TEAD4 recognized three bands in WT and showed that the two faster-migrating bands were decreased in *Tead1* cKO. Anti-TEAD2 recognized a band of ~65 kD in *Tead1* cKO that was barely detectable in WT lysates. This band corresponded to the size of TEAD2 reported in other cell types (*Beverdam et al., 2013*). We also found that the ~65 kD band was present early post-natally, but rapidly diminished with age (*Figure 1—figure supplement 1*), in accordance with the transcript data from mouse SCs (*Gerber et al., 2021*). Thus, the expression of TEADs 2–4 was altered in *Tead1* cKO, likely reflecting their expression in SCs, consistent with the single cell transcriptomic analysis of SCs (*Gerber et al., 2021*). Of these anti-TEAD antibodies, only anti-pan TEAD functioned

for immunofluorescence staining, verifying the presence of other TEADs in SC nuclei lacking TEAD1 (*Figure 6—figure supplement 1C and E*).

In the absence of a partner TF, nuclear YAP/TAZ are decreased and phosphorylation of YAP/TAZ increased, due to targeting by Large tumor suppressor kinase 1/2 (LATS1/2), negative regulators of YAP/TAZ (*Hao et al., 2008*; *Zhao et al., 2007*). To investigate if YAP/TAZ are in the nuclei of SCs and therefore remain transcriptionally functional in the absence of TEAD1, we compared Western blotting of P50 WT vs *Tead1* cKO SN lysates and observed an overall decrease in YAP levels in *Tead1* cKO but a >twofold increase in TAZ levels compared to WT (*Figure 1C*). This ratio of TAZ to YAP in *Tead1* cKO resembled that observed during developmental myelination in WT SNs (*Deng et al., 2017*; *Grove et al., 2017*). Furthermore, phosphorylation of YAP decreased in *Tead1* cKO, which would typically indicate increased YAP activity. Thus, YAP/TAZ are not targeted for destruction in the absence of TEAD1 in SCs. Indeed, YAP/TAZ were retained in the SC nucleus of *Tead1* cKO (*Figure 1D*), presumably due to other TEADs in the absence of TEAD1.

## TEAD1 is required for myelin formation and growth

*Tead1* cKO mice were smaller than WT mice at all ages, with labored breathing and widely splayed fore- and hindlimbs, indicative of severe peripheral neuropathy (*Figure 2A*, *Video 1*). Because of the neuropathy, *Tead1* cKO mice were humanely euthanized before P60. *Tead1* cKO SNs were noticeably translucent, indicative of reduced myelin content (*Figure 2B*). Electrical stimulation of cKO SNs elicited low and dispersed compound muscle action potentials (CMAPs; *Figure 2C*). Nerve conduction velocity (NCV) in cKO was ~8 m/s, which was markedly reduced from ~40 m/s in WT mice (*Figure 2D*).

Next, we used semi-thin and transmission electron microscopy (TEM) to examine the processes of developmental myelination and Remak bundle formation. We first examined SNs at P3, when, in WT, many large axons had already been radially sorted, establishing 1:1 relationship with promyelinating SCs, and myelination had begun. Compared to WT, *Tead1* cKO SNs were thinner, unsorted bundles of axons were larger and more numerous, and myelination was minimal (*Figure 2E*). Nevertheless, we frequently observed large axons that had been sorted and established a 1:1 relationship with promyelinating SCs, as in WT. Therefore, radial sorting was delayed, but not blocked, in *Tead1* cKO.

At P22, when WT SNs were fully myelinated, *Tead1* cKO SNs were largely unmyelinated and exhibited areas of unsorted axon bundles, of sorted but unmyelinated axons, and of thinly myelinated axons (*Figure 2F*, Figure 2—figure supplement 1A). Notably, at P50, this phenotype was unchanged from that at P22: areas of unsorted bundles, promyelination, and thin myelination persisted (*Figure 2G*, *Figure 2—figure supplement 1A*). TEM analysis confirmed that myelin was indeed thinner in *Tead1* cKO than in WT at both P22 and P50 (*Figure 2F and G*, *Figure 2—figure supplement 1B*). The myelin thickness measured as g-ratio, that is, the ratio between the axon diameter and the fiber diameter (axon plus myelin sheath), did not significantly change in cKO between P22 and P50 (*Figure 2H*), suggesting that growth of the myelin sheath relative to axon diameter did not increase over this period. These data suggest that myelination in *Tead1* cKO was arrested rather than delayed. Plots of g-ratio versus axon diameter showed that the mutant defect was present in axons of all diameters but was particularly notable in those of larger diameter (*Figures 2I and 3J*). The mean axon diameter was also reduced in cKO at P22, although it increased with age (*Figure 2K*). Collectively, these data show that TEAD1 is required for timely radial sorting, myelin formation and growth during development. Although some axons become myelinated in *Tead1* cKO, their myelin sheaths fail to properly grow in proportion to axon diameter in the absence of TEAD1.

## TEAD1 is required for proper formation of Remak bundles

Remak bundle formation was also abnormal in *Tead1* cKO mice. In WT SNs at P22 and P50, small diameter axons were present in Remak bundles, with each C-fiber ensheathed by SC cytoplasm (*Figure 2F and L*). In *Tead1* cKO, C-fibers were also present in bundles in which axons were rarely larger than 1 µm in diameter, suggesting that small diameter axons were almost fully sorted. However, at both P22 and P50, the observed bundles were frequently small, often the size of a large myelinated axon, and were occasionally thinly myelinated. Notably, a SC surrounded each bundle but individual C-fibers of most Remak bundles were not ensheathed by SC cytoplasm (*Figure 2L*). Quantification showed that ~80% of C-fibers were not ensheathed by SC cytoplasm in P50 *Tead1* cKO SNs (*Figure 2M*). Thus,

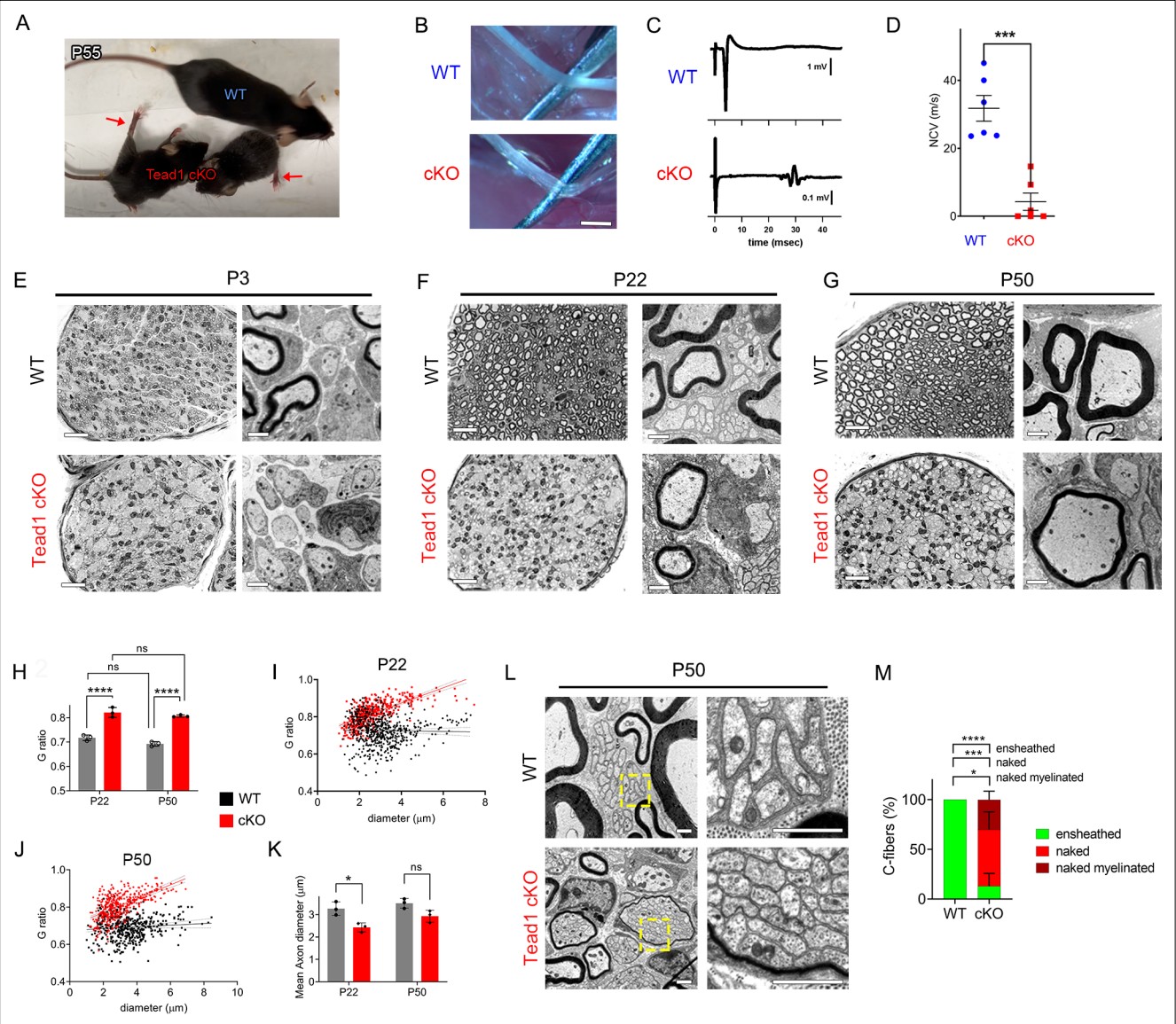

**Figure 2.** TEAD1 is required for myelin sheath formation and growth. (**A**) A living control (WT) and two *Tead1* cKO mice from one litter at P55. Arrows point to splayed paralyzed hindlimbs. (**B**) Exposed SNs of WT and *Tead1* cKO mice during dissection. Scale bar: 1 mm (**C**) Representative images of CMAPs generated by stimulation of WT and *Tead1* cKO Sns. (**D**) Quantification of NCV in WT and *Tead1* cKO. n=6 mice per genotype. ***p<0.001, Student's unpaired t-test. (**E**) Representative images of semithin (left panels) and TEM (right panels) of transverse sections of WT and *Tead1* cKO SNs at P3. Scale bar: 20 μm (left panels), 1 μm (right panels) (**F**) Representative images of semithin (left panels) and TEM (right panels) of transverse sections of WT and *Tead1* cKO SNs at P22. Scale bar: 20 μm (left panels), 1 μm (right panels) (**G**) Representative images of semithin (left panels) and TEM (right panels) of transverse sections of WT and *Tead1* cKO SNs at P50. Scale bar: 20 μm (left panels), 1 μm (right panels) (**H**) Quantification of G-ratio for axons in WT and *Tead1* cKO SNs at P22 and at P50. n=3 mice per group. ns, p>0.05 (P22 WT versus P50 WT, P22 cKO versus P50 cKO), ****p<0.0001 (P22 WT versus P22 cKO, P50 WT versus P50 cKO). (**I**) Scatter plot graph displaying G-ratio in relation to axon diameter in WT and *Tead1* cKO SNs at P22. n=1063 axons from 3 mice per each genotype. (**J**) Scatter plot graph displaying G-ratio in relation to axon diameter in WT and *Tead1* cKO SNs at P50. n=880 axons from 3 mice per each genotype. (**K**) Quantitative comparison of mean axon diameter in WT and *Tead1* cKO SNs at P22 and at P50. n=3 mice per group. ns, p>0.05, *p≤0.05, two-way ANOVA with Tukey's multiple comparisons test. (**L**) Representative TEM images of transverse sections of P50 WT and *Tead1* cKO SNs. Right panels are enlarged images of the boxed area in left panels. Scale bar: 1 μm (**M**) Quantification of normal and abnormal C-fibers in P50 WT and *Tead1* cKO SNs. C-fibers normally ensheathed by SC cytoplasm are marked as 'ensheathed', whereas those not ensheathed are denoted as 'naked'. Naked C-fibers located in a myelinated bundle are denoted as 'naked myelinated'. Relative percentage of each type of C-fiber is plotted. N>500 C-fibers from 3 mice per each genotype. *p≤0.05, ***p≤0.001, ****p<0.0001, two-way ANOVA Tukey's multiple comparisons test.

The online version of this article includes the following source data and figure supplement(s) for figure 2:

**Source data 1.** This zip archive contains source files for graphs in *Figure 2D, H, I, J, K and M*.

**Figure supplement 1.** Additional semi-thin and TEM images of WT and *Tead1* cKO.

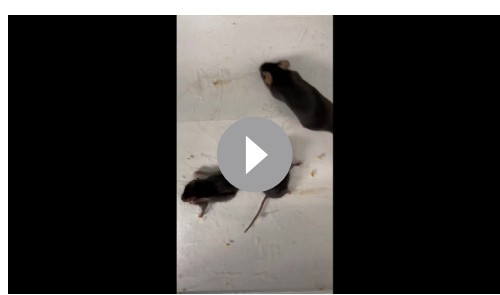

**Video 1.** A movie showing a P55 WT and two littermates *Tead1* cKO mic.
https://elifesciences.org/articles/87394/figures#video1

TEAD1 is required to develop proper association of C-fibers with non-myelinating SCs in Remak bundles.

## TEAD1 regulates SC proliferation both positively and negatively to promote myelination

For myelination and Remak bundle formation to occur properly, timely generation of sufficient numbers of SCs is essential. To determine if the defects of myelination and Remak bundles in *Tead1* cKO are due to defective SC proliferation, we counted SCs. At P3, SC number in cKO SNs was about half that of WT (*Figure 3A*). Consistently, the number of SCs in the cell cycle (Ki67-positive; *Figure 3B*) and in S-phase (EdU-positive; *Figure 3C*) was greatly reduced in cKO at P3. Thus, TEAD1 is required for optimal SC proliferation during development.

Notably, at P22 and P50, SC numbers in cKO were markedly increased and became more than twice those of WT (*Figure 3A*). As cKO SC numbers were similar at P22 and P50, we reasoned that between P3 and P22, SC proliferation must occur at an abnormally high rate. Indeed, the number of SCs that were Ki67-positive or EdU-positive was significantly increased in cKO at P22 (*Figure 3B and C*). The increased SC numbers were not due to proliferation of residual SCs that express TEAD1: TEAD1 was absent in most EdU-positive SCs in *Tead1* cKO (*Figure 3D*). These data suggest that, during development, SCs switch from requiring TEAD1 for optimal proliferation, to requiring TEAD1 for timely exit from the cell-cycle, which is a prerequisite for SC differentiation and the onset of myelination (*Hindley and Philpott, 2012*).

## TEAD1 is required for Krox20 to upregulate myelin proteins, but not for Krox20 upregulation

Our findings thus far show that, in the absence of TEAD1 in the SCs of *Tead1* cKO, most axons are not myelinated, which grossly phenocopied *Yap1/Wwtr1* iDKO (*Yap1* and *Wwtr1* encode the proteins commonly listed as YAP and TAZ, respectively). *Yap1/Wwtr1* cDKO exhibited a complete failure of myelination due to arrest of their SCs as promyelinating SCs, in part because they fail to upregulate Krox20/Egr2 (Early Growth Response 2), which drives transcription of myelin proteins (*Grove et al., 2017*). To determine whether TEAD1 is required for Krox20 upregulation, we first investigated Oct6 (Pou3f1, Scip), whose upregulation precedes and is required for Krox20 upregulation; Oct6 is subsequently downregulated, possibly by Krox20 in a feedback mechanism (*Ryu et al., 2007*; *Zorick et al., 1999*). Western blotting of P50 SN lysates showed minimal expression of Oct6 in WT, but abnormally upregulated expression in *Tead1* cKO (*Figure 4A*). Immunohistochemistry of P50 cKO SNs also showed intense Oct6 expression in ~80% of SC nuclei; Oct6 expression was minimal in WT (*Figure 4B*). Notably, at both P11 and P50, a similar percentage of SCs expressed Krox20 in *Tead1* cKO, as in WT SNs (*Figure 4C*). In Western blots, Krox20 was upregulated >fourfold in cKO compared to WT (*Figure 4A*). As adult *Tead1* cKO mice contain ~twofold more SCs than WT (*Figure 3A*), these data suggest that each SC in *Tead1* cKO contains ~twofold more Krox20 than WT. Thus, in contrast to SCs lacking YAP/TAZ, SCs lacking TEAD1 upregulate Krox20, suggesting that TEAD1 is dispensable for YAP/TAZ to upregulate Krox20.

Consistent with the idea that Krox20 is pivotal for myelin gene transcription, expression of myelin basic protein (MBP), and myelin protein zero (MPZ), was abundant in WT P50 SNs (*Figure 4A*). In contrast, despite ample expression of Krox20, both MBP and MPZ were markedly decreased in *Tead1* cKO, indicating that Krox20 fails to properly upregulate myelin proteins without TEAD1. To exclude the possibility that Krox20 expressed in cKO SCs is defective, we took advantage of the earlier finding that loss of Krox20 or its co-factors, Nab1/2, causes SCs to remain aberrantly in the cell-cycle (*Le et al., 2005b*; *Zorick et al., 1999*). Therefore, we studied whether EdU-positive SCs in the cell-cycle of P11 cKO were also positive for Krox20, which would indicate defective Krox20. We found that, although ~65% of SC nuclei contained Krox20, only a few EdU-positive SCs exhibited even minimal

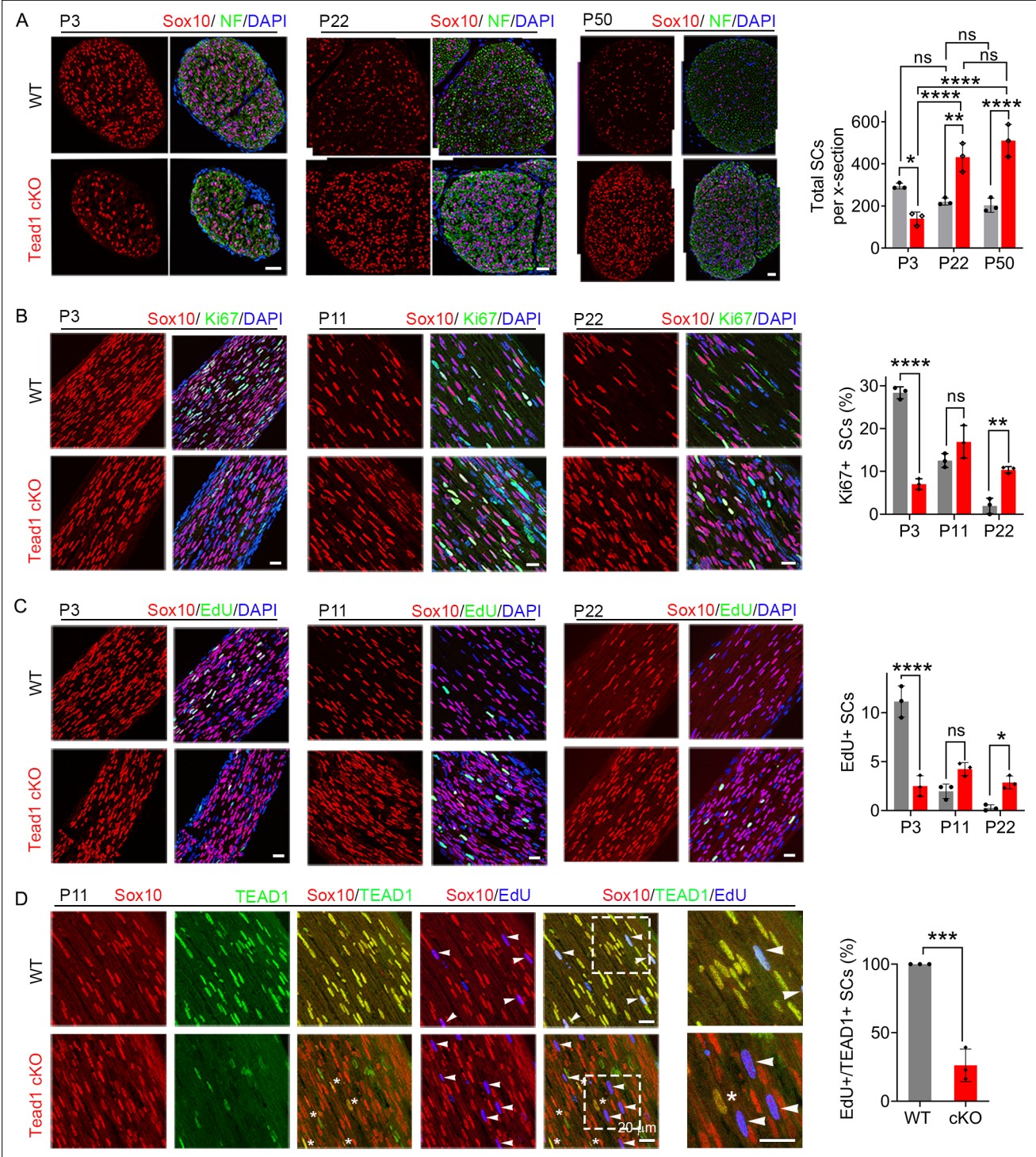

**Figure 3.** TEAD1 regulates SC proliferation both positively and negatively. (**A**) Immunostaining and quantification of SCs on transverse sections of WT and *Tead1* cKO SNs at P3, P22, and P52. SC nuclei and all cell nuclei are marked by Sox10 and DAPI. Axons are marked by Neurofilaments (NF). n=3 mice per genotype, ns, p>0.05, *p≤0.05, **p≤0.01, ****p<0.0001, two-way ANOVA with Tukey's multiple comparisons test. Scale bar: 30 µm (**B**) Immunostaining and quantification of Ki67 +SCs on longitudinal sections of WT and *Tead1* cKO SNs at P3, P11, and P22. SC nuclei and all cell nuclei are marked by Sox10 and DAPI. n=3 mice per genotype, ns, p>0.05, **p≤0.01, ****p<0.0001, two-way ANOVA with Tukey's multiple comparisons test. Scale bar: 30 µm (P3 panels), 20 µm (P11 and P22 panels) (**C**) Immunostaining and quantification of EdU +SCs on longitudinal sections of WT and *Tead1* cKO SNs at P3, P11, and P22. SC nuclei and all cell nuclei are marked by Sox10 and DAPI. n=3 mice per genotype, ns, p>0.05, *p≤0.05, ****p<0.0001, two-way ANOVA with Tukey's multiple comparisons test. Scale bar: 30 µm (P3 panels), 20 µm (P11 and P22 panels) (**D**) Immunostaining and quantification of EdU+/TEAD1+SCs on longitudinal sections of WT and *Tead1* cKO SNs at P11. SC nuclei are marked by Sox10. Asterisks denote *Tead1* cKO SCs in

*Figure 3 continued on next page*

Figure 3 continued

which TEAD1 is not deleted and which are not EdU+. Arrowheads denote EdU +SCs which are TEAD1+in WT but TEAD1- in *Tead1* cKO. n=3 mice per genotype, ***p<0.001, Student's unpaired t-test. Scale bar:20 µm.

The online version of this article includes the following source data for figure 3:

**Source data 1.** This zip archive contains source files for graphs in *Figure 3A, B, C and D*.

---

Krox20 immunoreactivity in either WT or *Tead1* cKO (*Figure 4D*). Thus, despite high levels of functional Krox20, myelin protein expression was greatly reduced in cKO SCs, suggesting that TEAD1 is dispensable for Krox20 upregulation but required for Krox20 to upregulate transcription of myelin proteins during development.

## TEAD1 and YAP/TAZ upregulate cholesterol biosynthetic enzymes, FDPS and IDI1

Myelination of peripheral nerves by SCs requires a large amount of lipid and cholesterol biosynthesis. De novo biosynthesis of fatty acids (integral building blocks of lipids) and cholesterol is crucially important for SCs to initiate myelination and to grow a myelin sheath (*Montani et al., 2018*; *Norrmén et al., 2014*; *Saher et al., 2009*; *Verheijen et al., 2009*). YAP/TAZ may regulate these pathways, as suggested by reduced mRNA expression in *Yap1/Wwtr1* cDKO of SREBPs TFs (Sterol regulatory element binding proteins) and their target enzymes involved in lipid/cholesterol biosynthesis (*Deng et al., 2017*; *Poitelon et al., 2016*). We therefore investigated, at the protein level, whether TEAD1 and YAP/TAZ are required for fatty acid and/or cholesterol synthesis. To test this premise, we examined expression of SREBP1 and its target SCD1 (Stearoyl-CoA desaturase-1), which upregulates fatty acid synthesis, and of SREBP2 and its targets HMGCR (HMG-CoA reductase), FDPS (Farnesyl Diphosphate Synthase) and IDI1 (Isopentenyl-diphosphate delta isomerase 1), which upregulate cholesterol synthesis (*Leblanc et al., 2005*; *Verheijen et al., 2009*). Because SREBP pathways are upregulated early postnatally for myelination in WT, their upregulation may simply be delayed in developmentally delayed *Tead1* cKO. We therefore investigated SREBP pathways both at P8 and P40 in WT and *Tead1* cKO mice. Western blot analysis showed that levels of nuclear SREBP1 were not affected by loss of TEAD1 at P8, and only slightly reduced at P40 (*Figure 5A*). SCD1 was reduced at P8, but slightly increased at P40. In contrast to SREBP1, SREBP2 was ~50% reduced in *Tead1* cKO at both P8 and P40. Despite marked reduction in SREBP2, HMGCR was not reduced at P8 or P40. However, both FDPS and IDI1 levels were greatly reduced at both P8 and P40 (*Figure 5A*).

It is possible that HMGCR was not reduced in cKO due to TEAD1 redundancy and compensation by other TEADs. If so, then HMGCR levels should be reduced in *Yap1/Wwtr1* cDKO, because YAP/TAZ are the primary partners for all TEADs (*Holden and Cunningham, 2018*; *Zheng and Pan, 2019*). To test this idea, we examined SREBP pathways in *Yap1/Wwtr1* cDKO SNs at P60 because developmental delay is greater in *Yap1/Wwtr1* cDKO than in *Tead1* cKO SNs (*Grove et al., 2017*). We found that, as in *Tead1* cKO, SREBP1 levels were unaffected and SCD1 levels were markedly increased by the loss of YAP/TAZ in cDKO at P60 (*Figure 5B*). Unlike in *Tead1* cKO, levels of SREBP2 were not markedly reduced in Yap1/Wwtr1 cDKO. As in *Tead1* cKO, however, HMGCR levels were not reduced in Yap1/Wwtr1 cDKO, whereas levels of FDPS and IDI1 were greatly reduced. These results therefore indicate that YAP/TAZ-TEAD1 transcriptional complex selectively upregulates IDI1 and FDPS, contributing to cholesterol biosynthesis. Together with the lack of synthesis of myelin proteins, this reduction in cholesterol biosynthesis may prevent myelin formation and growth in *Tead1* cKO.

## TEAD1 is largely dispensable for myelin maintenance in adult *Schwann cells*

Because TEAD1 is required for myelin formation and growth and Remak bundle formation during development, we next investigated its role in maintenance and regeneration of peripheral nerves in adults. We crossed *Tead1*^fl/fl mice with mice carrying a tamoxifen-inducible *Plp1*-CreER (*Leone et al., 2003*) to generate *Plp1* -CreER; *Tead1*^fl/fl (hereafter *Tead1* iKO) and inducibly inactivated SC *Tead1* in adult *Tead1* iKO mice. Western blotting showed markedly reduced TEAD1 in *Tead1* iKO (*Figure 6— figure supplement 1A*). Residual TEAD1 is likely not associated with SCs. Indeed, immunostaining of WT and induced iKO SNs consistently showed that TEAD1 was deleted in >95% of SC nuclei in

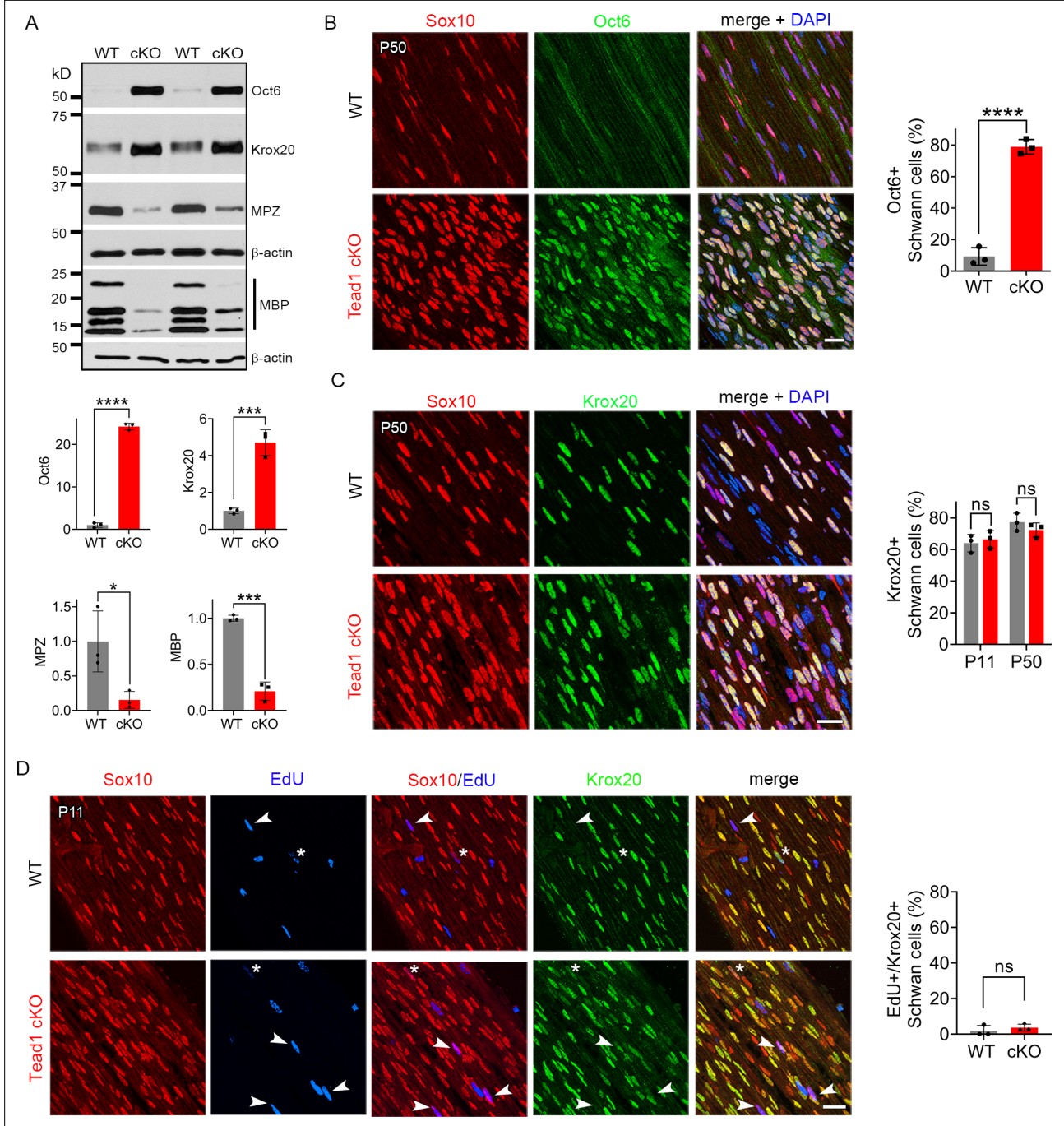

**Figure 4.** TEAD1 is required for Krox20 to upregulate myelin proteins during development. (**A**) Western blotting of P50 WT and *Tead1* cKO SN lysates with anti-Oct6, -Krox20, -MPZ, and -MBP. Protein expression is normalized to that of *β-actin* as an internal control, and WT expression is arbitrarily given the value 1. n=3 mice per genotype. *p≤0.05, ***p≤0.001, ***p≤0.0001, Student's unpaired t-test. (**B**) Immunostaining and quantification of Oct6 +SCs on longitudinal sections of P50 WT and *Tead1* cKO SNs. SC nuclei are identified by Sox10. All cell nuclei are marked by DAPI. n=3 mice per genotype, ****p<0.0001, Student's unpaired t-test. Scale bar: 20 µm (**C**) Immunostaining and quantification of Krox20 +SCs on longitudinal sections of WT and *Tead1* cKO SNs at P11 and P50. SC nuclei are identified by Sox10. All cell nuclei are marked by DAPI. n=3 mice per each group, ns, p>0.05, Student's unpaired t-test. Scale bar: 20 µm (**D**) Immunostaining and quantification of EdU+/Krox20 +SCs on longitudinal sections of WT and *Tead1* cKO SNs at P11. SC nuclei are identified by Sox10. Arrowheads denote examples of EdU +SCs which are Krox20-. Asterisks denote rarely observed EdU +SCs which are Krox20+. n=3 mice per genotype, ns, p>0.05, Student's unpaired t-test. Scale bar: 20 µm.

The online version of this article includes the following source data for figure 4:

**Source data 1.** This zip archive contains source files for graphs in *Figure 4A, B, C and D* and uncropped labeled or unlabeled blots of *Figure 4A*.

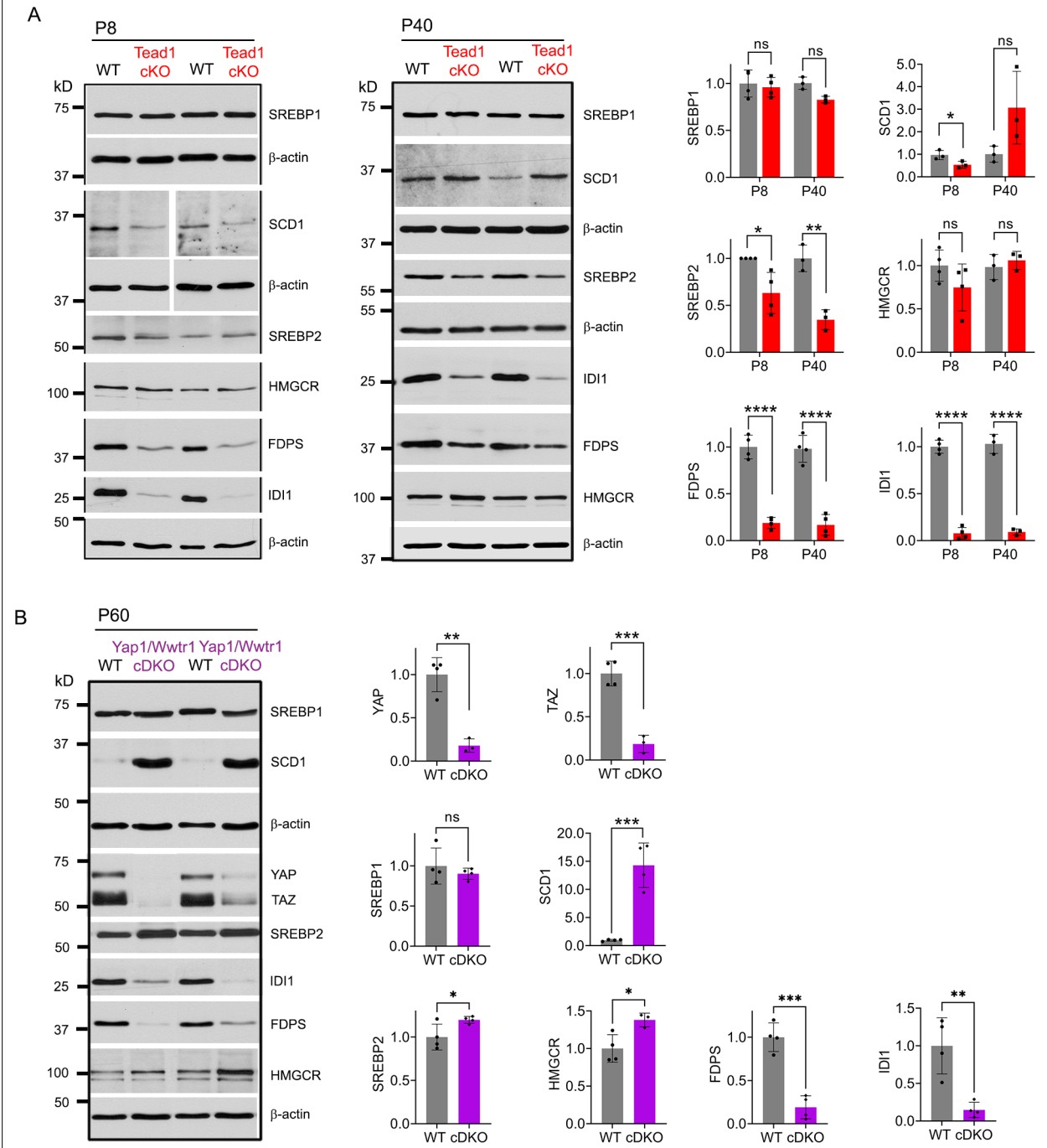

**Figure 5.** TEAD1 and YAP/TAZ upregulate cholesterol synthesis enzymes: FDPS and IDI1. (**A**) Western blotting of P8 and P40 WT and *Tead1* cKO SN lysates with anti-SREBP1, -SCD1, -SREBP2, -HMGCR, -FDPS, and -IDI1. Protein expression is normalized to that of *β-actin* as an internal control, and WT expression is arbitrarily given the value 1. n=3 or 4 mice per genotype. ns, p>0.05, *p≤0.05, **p≤0.01, ****p<0.0001, Student's unpaired t-test. (**B**) Western blotting of P60 WT and *Yap1/Wwtr1* cDKO SN lysates with anti-YAP, -TAZ, -SREBP1, -SCD1, -SREBP2, -HMGCR, -FDPS, and -IDI1. Protein expression is normalized to that of *β-actin* as an internal control, and WT expression is arbitrarily given the value 1. n=3 or 4 mice per genotype. ns, p>0.05, *p≤0.05, **p≤0.01, ***p<0.001, Student's unpaired t-test.

The online version of this article includes the following source data for figure 5:

**Source data 1.** This zip archive contains uncropped labeled or unlabeled blots of P8 WT and Tead1 cKO SNs in *Figure 5A*.

**Source data 2.** This zip archive contains source files for graphs in *Figure 5A and B*, and uncropped labeled or unlabeled blots of P40 WT and Tead1 cKO SNs in *Figure 5A* and P60 SNs of WT and YAP/TAZ cDKO in *Figure 5B*.

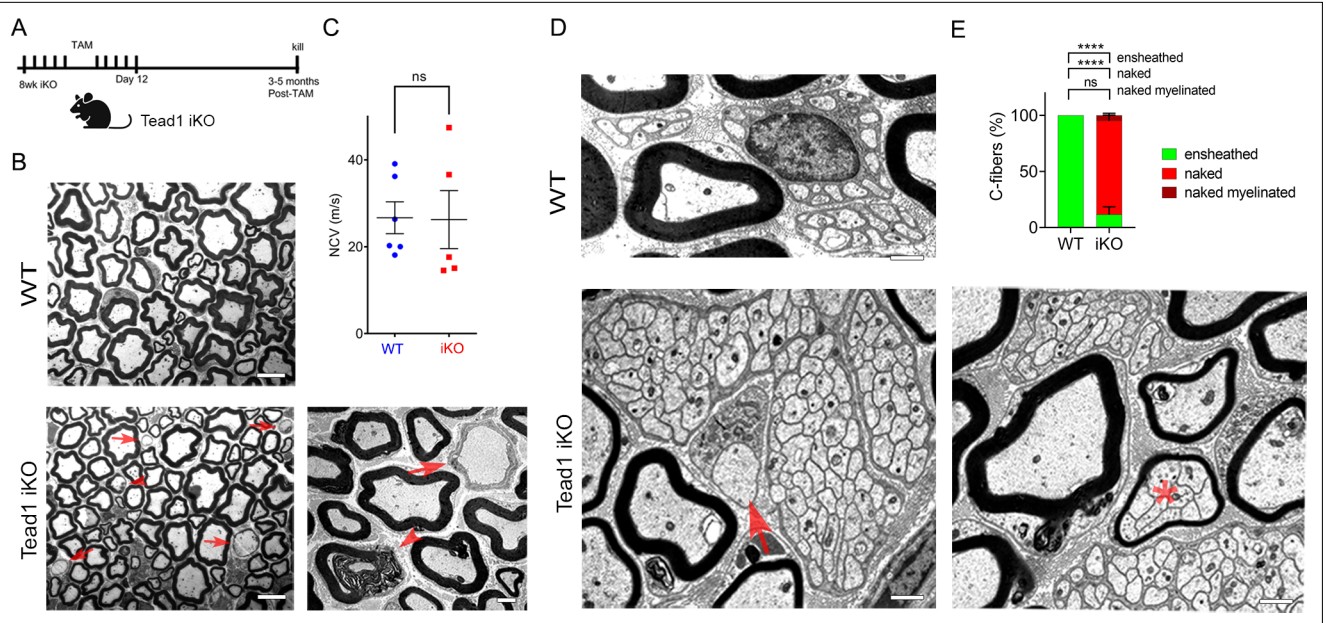

**Figure 6.** TEAD1 is largely dispensable for myelin maintenance, but required for Remak bundle integrity. (**A**) Cartoon showing timeline of tamoxifen (TAM) injection and time of sacrifice of *Tead1* iKO mice. (**B**) Representative TEM images of P150 WT and *Tead1* iKO SNs showing occasional large axons undergoing demyelination (arrowheads) or completely demyelinated (arrows). Scale bar: 6 µm (left panels), 2 µm (right panel) (**C**) Quantification of NCV in P90-P150 WT and *Tead1* iKO mice. n=5 or 6 mice per genotype. ns, p>0.05, Student's unpaired t-test. (**D**) Representative TEM images of P97 WT and *Tead1* iKO SNs showing Remak bundles. Asterisk denotes a bundle of thinly myelinated C-fibers. Arrow denotes demyelinated axon. Scale bar: 1 µm (**E**) Quantification of normal and abnormal C-fibers in P150 WT and *Tead1* iKO SNs. Relative percentage of each type of C-fiber is plotted. N>500 C-fibers from 3 mice per each genotype. ns, p>0.05, ****p<0.0001, two-way ANOVA with Tukey's multiple comparisons test.

The online version of this article includes the following source data and figure supplement(s) for figure 6:

**Source data 1.** This zip archive contains source files for graphs in *Figure 6C and E*.

**Source data 2.** This zip archive contains source files for graphs in *Figure 6A–G* and uncropped labeled or unlabeled blots of *Figure 6A*.

**Figure supplement 1.** Expression of TEAD1, pan-TEAD and YAP/TAZ in intact and regenerating nerves of WT and *Tead1* iKO mice.

both intact and regenerating iKO SNs, whereas TEAD1 was present in 100% of SC nuclei of WT SNs (*Figure 6—figure supplement 1B and D*). As anticipated, iKO SC nuclei in both intact and regenerating nerves were strongly immunostained by pan-TEAD antibody, presumably indicating the presence of other TEADs (*Figure 6—figure supplement 1C and E*). We also found that YAP/TAZ were transcriptionally functional in adult SCs lacking TEAD1, as indicated by strong nuclear expression of YAP/TAZ in SCs of both intact and regenerated iKO SNs (*Figure 6—figure supplement 1F and G*).

*Tead1* iKO mice appeared normal without obvious signs of peripheral neuropathy up to 5 months post-tamoxifen, the longest time of observation (*Figure 6A*). TEM analysis of iKO SNs between 3 and 5 months post-tamoxifen revealed, however, some large axons undergoing demyelination or completely demyelinated, which were undetectable in WT SNs (*Figure 6B*). The demyelinated axons were observed in discrete regions of SNs, indicating segmental demyelination (<5% large axons on 12 TEM sections exhibiting demyelinated axons). NCV of these iKO mice did not significantly differ from that of WT (*Figure 6C*). Therefore, loss of TEAD1 in adult SCs evoked only modest focal demyelination in a small number of axons, which did not cause significant functional impairment of peripheral nerves.

## TEAD1 is required for maintaining structural integrity of Remak bundles

Next, we examined Remak bundles in *Tead1* iKO mice. As in WT SNs, C-fibers in WT mice were individually ensheathed by SC cytoplasm. In stark contrast, in *Tead1* iKO, many C-fibers in most Remak bundles were not ensheathed by SC cytoplasm (*Figure 6D*). Furthermore, some small Remak bundles were occasionally myelinated (*Figure 6E*), as in *Tead1* cKO SNs. Thus, TEAD1 is required for formation as well as maintenance of Remak bundles.

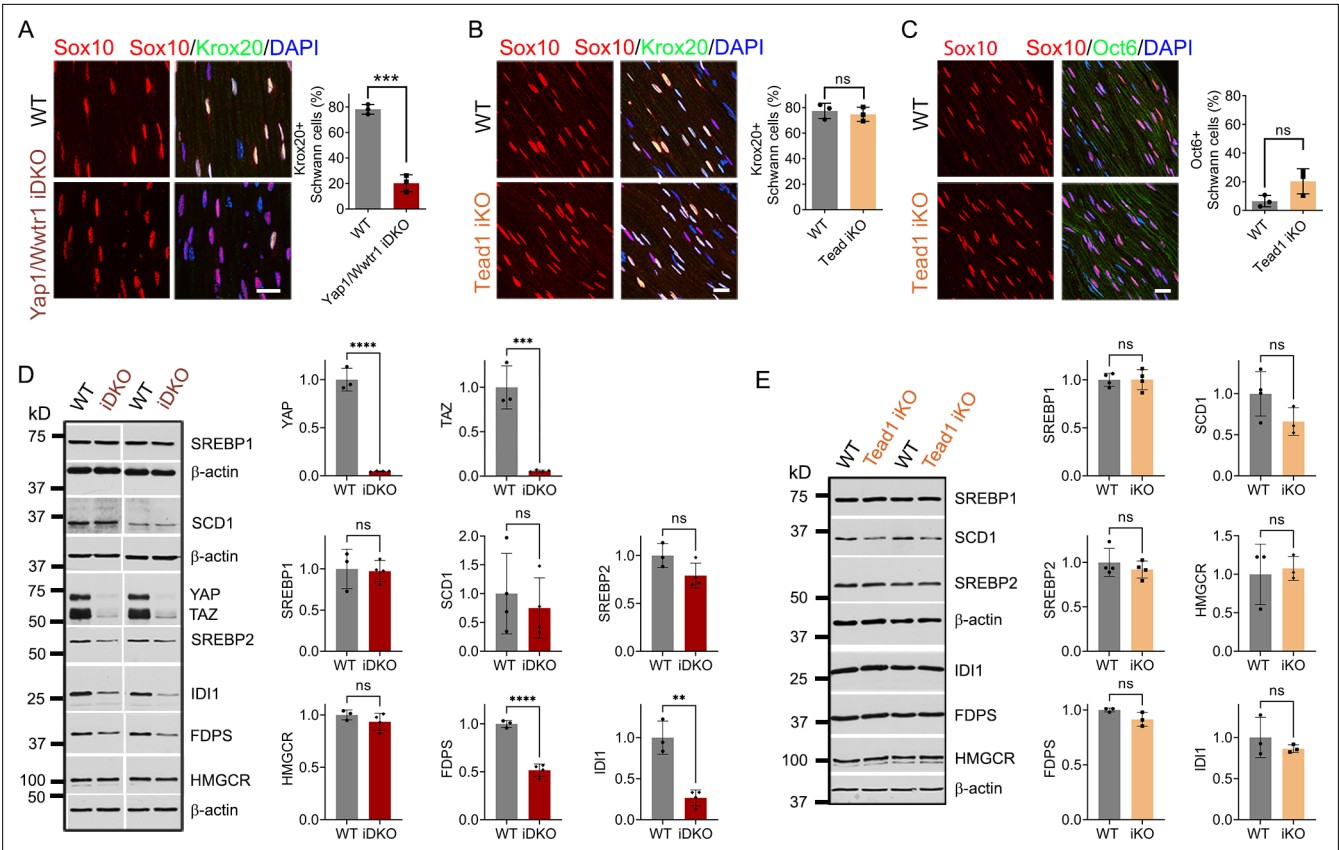

**Figure 7.** TEAD1 is dispensable for YAP/TAZ to maintain Krox20, FDP and IDI1 expression. (**A**) Immunostaining and quantification of Krox20 +SCs on longitudinal sections of WT and *Yap1/Wwtr1* iDKO SNs 2 weeks after 1st tamoxifen injection. SC nuclei are marked by Sox10. All cell nuclei are marked by DAPI. n=3 mice per genotype, ***p<0.001, Student's unpaired t-test. Scale bar: 20 µm (**B**) Immunostaining and quantification of Krox20 +SCs on longitudinal sections of WT and *Tead1* iKO SNs 6 weeks after 1st tamoxifen injection. SC nuclei are marked by Sox10. All cell nuclei are marked by DAPI. n=3 mice per genotype, ns, p>0.05, Student's unpaired t-test. Scale bar: 20 µm (**C**) Immunostaining and quantification of Oct6 +SCs on longitudinal sections of WT and *Tead1* iKO SNs 6 weeks after 1st tamoxifen injection. SC nuclei are marked by Sox10. All cell nuclei are marked by DAPI. n=3 mice per genotype, ns, p>0.05, Student's unpaired t-test. Scale bar: 20 µm (**D**) Western blotting of WT and *Yap1/Wwtr1* iDKO SN lysates with anti-YAP, -TAZ, -SREBP1, -SCD1, -SREBP2, -HMGCR, -FDPS, and -IDI1. Protein expression is normalized to that of *β-actin* as an internal control, and WT expression is arbitrarily given the value 1. n=3 or 4 mice per genotype. ns, p>0.05, **p≤0.01, ***p<0.001, ****p<0.0001, Student's unpaired t-test. (**E**) Western blotting of WT and *Tead1* iKO SN lysates with anti-SREBP1, -SCD1, -SREBP2, -HMGCR, -FDPS, and -IDI1. Protein expression is normalized to that of *β-actin* as an internal control, and WT expression is arbitrarily given the value 1. n=3 or 4 mice per genotype. ns, p>0.05, Student's unpaired t-test.

The online version of this article includes the following source data for figure 7:

**Source data 1.** This zip archive contains source files for graphs in *Figure 7A–E* and uncropped labeled or unlabeled blots of *Figure 7D and E*.

## TEAD1 is dispensable for YAP/TAZ to maintain Krox20, FDP, and IDI1 expression

To investigate why the demyelinating phenotype of *Tead1* iKO is much milder than that of *Yap1/Wwtr1* iDKO, we first investigated expression of Krox20, which is required for myelin maintenance (*Decker et al., 2006*). We previously observed that Krox20 is markedly downregulated in *Yap1/Wwtr1* iDKO SCs, which correlated with frequent segmental demyelination and markedly reduced NCV (*Grove et al., 2017*). As a control, we injected *Plp1-CreER; Yap1^fl/fl; Wwtr1^fl/fl* mice with tamoxifen and reproduced our previous result: Krox20 was dramatically downregulated in most *Yap1/Wwtr1* iDKO SCs only 2 weeks post-tamoxifen (*Figure 7A*). In contrast, in *Tead1* iKO, Krox20 expression was indistinguishable from WT even at 6 weeks post-tamoxifen (*Figure 7B*). We then investigated Oct6 expression. Unlike in *Yap1/Wwtr1* iDKO SNs (*Grove et al., 2017*), Oct6 was not elevated in *Tead1* iKO SNs (*Figure 7C*). Thus, TEAD1 is dispensable for maintaining Krox20 expression in adult SCs.

Next, we investigated SREBP pathways, which are required for developmental myelination (*Saher et al., 2009; Verheijen et al., 2009*). Whether these pathways are also involved in myelin maintenance

has not been studied. We first assessed the effects of acute loss of YAP/TAZ on these pathways in adult SCs. Western blotting of WT and *Yap1/Wwtr1* iDKO SN lysates showed that the SREBP1 pathway and expression of SREBP2 and HMGCR in the SREBP2 pathway were unaffected in *Yap1/Wwtr1* iDKO (*Figure 7D*). However, FDPS and IDI1 levels were significantly downregulated in *Yap1/Wwtr1* iDKO, recapitulating the effect of developmental loss of YAP/TAZ on these proteins (*Figure 5B*). Thus, the acute downregulation of these key enzymes for cholesterol biosynthesis correlates with the acute demyelination previously observed in *Yap1/Wwtr1* iDKO mice. This result suggests that YAP/TAZ maintain myelination through both myelin gene expression and the cholesterol biosynthesis pathway.

Western blotting of *Tead1* iKO SN lysates revealed that the SREBP pathways were not significantly affected by loss of TEAD1 (*Figure 7E*), unlike *Tead1* cKO in which FDPS and IDI1 were markedly downregulated (*Figure 5A*). This result suggests that TEAD1 is dispensable for cholesterol synthesis in adult SCs.

## TEAD1 is absolutely required for myelination of regenerating axons after nerve injury

Next, we investigated whether TEAD1 is required for axon regeneration and remyelination after nerve injury. YAP/TAZ are not required for axon regeneration but are crucial for remyelination (*Grove et al., 2017*; *Jeanette et al., 2021*). Because *Yap1/Wwtr1* iDKO mice die ~2 weeks post-injury (wpi), it remains possible that remyelination is delayed rather than completely blocked. In contrast, *Tead1* iKO mice are not susceptible to early death. We, therefore, examined them 4 weeks after crushing SN (*Figure 8A*), when hindlimb muscles are fully reinnervated in WT. We first verified efficient deletion of TEAD1 in >95% SCs of *Tead1* iKO mice after nerve crush (*Figure 6—figure supplement 1D*). Longitudinal sections of iKO SNs showed numerous Tuj1-labeled axons that regenerated along the distal nerve stump, as in WT SNs (*Figure 8B*). However, MBP immunoreactivity was greatly reduced in the distal nerve stump of *Tead1* iKO, compared to WT, suggesting that axons regenerated but were not remyelinated. We also found that regenerated axons reached hindlimb muscles and reinnervated neuromuscular junctions (NMJs) in iKO, as in WT (*Figure 8C*). Moreover, *Tead1* iKO frequently exhibited features of paralyzed muscles, such as axon terminals that grew past the synaptic area, fragmented clusters of acetylcholine receptors (AChRs), and satellite extra-synaptic AChRs (*Wright et al., 2007*). Semithin and EM analysis confirmed virtual absence of myelinated axons in iKO SNs; although large axons were associated with SCs, they were not myelinated (*Figure 8D*). Accordingly, electrical stimulation of iKO SNs did not elicit CMAPs or NCV (*Figure 8E*). Thus, like YAP/TAZ, TEAD1 in SCs is not required for axon regeneration but is required for remyelination after nerve injury.

After injury, SCs transdifferentiate into growth-promoting repair SCs, which proliferate and reach a peak around 3 days post-injury (dpi) (*Clemence et al., 1989*; *Jessen and Mirsky, 2016*; *Tricaud and Park, 2017*). Quantitative analysis of SC proliferation at 3 dpi by counting EdU$^+$/Sox10$^+$ SCs showed that a similar percentage of SCs were in S-phase in both WT and *Tead1* iKO SNs (*Figure 8F*), suggesting that, like YAP/TAZ (*Grove et al., 2017*), TEAD1 is not required for proliferation of repair SCs. Lastly, we counted Krox20$^+$ and Oct6$^+$ SCs at 4 wpi. Notably, in contrast to *Yap1/Wwtr1* iDKO SCs, which fail to upregulate Krox20 after injury (*Grove et al., 2017*), Krox20 was upregulated in *Tead1* iKO SCs, as in WT SCs (*Figure 8G*). Furthermore, Oct6 was aberrantly elevated in *Tead1* iKO SCs, like in *Tead1* cKO SCs during development (*Figure 8H*). These results suggest that, similar to developmental myelination, remyelination requires TEAD1 for Oct6 downregulation and for Krox20 to upregulate myelin proteins. Additionally, almost complete absence of remyelination at 4 wpi highlights the non-redundant, crucial role of TEAD1 in the remyelination of regenerating nerves after injury.

## Discussion

The present study conclusively establishes TEAD1 as a crucial TF mediating myelin formation, growth, and regeneration. TEAD1 promotes myelination by dually regulating SC proliferation, by enabling induced Krox20 to upregulate myelin proteins, and by upregulating cholesterol biosynthetic enzymes. Our findings also suggest that TEAD1 is critical in the proper formation and maintenance of Remak bundles.

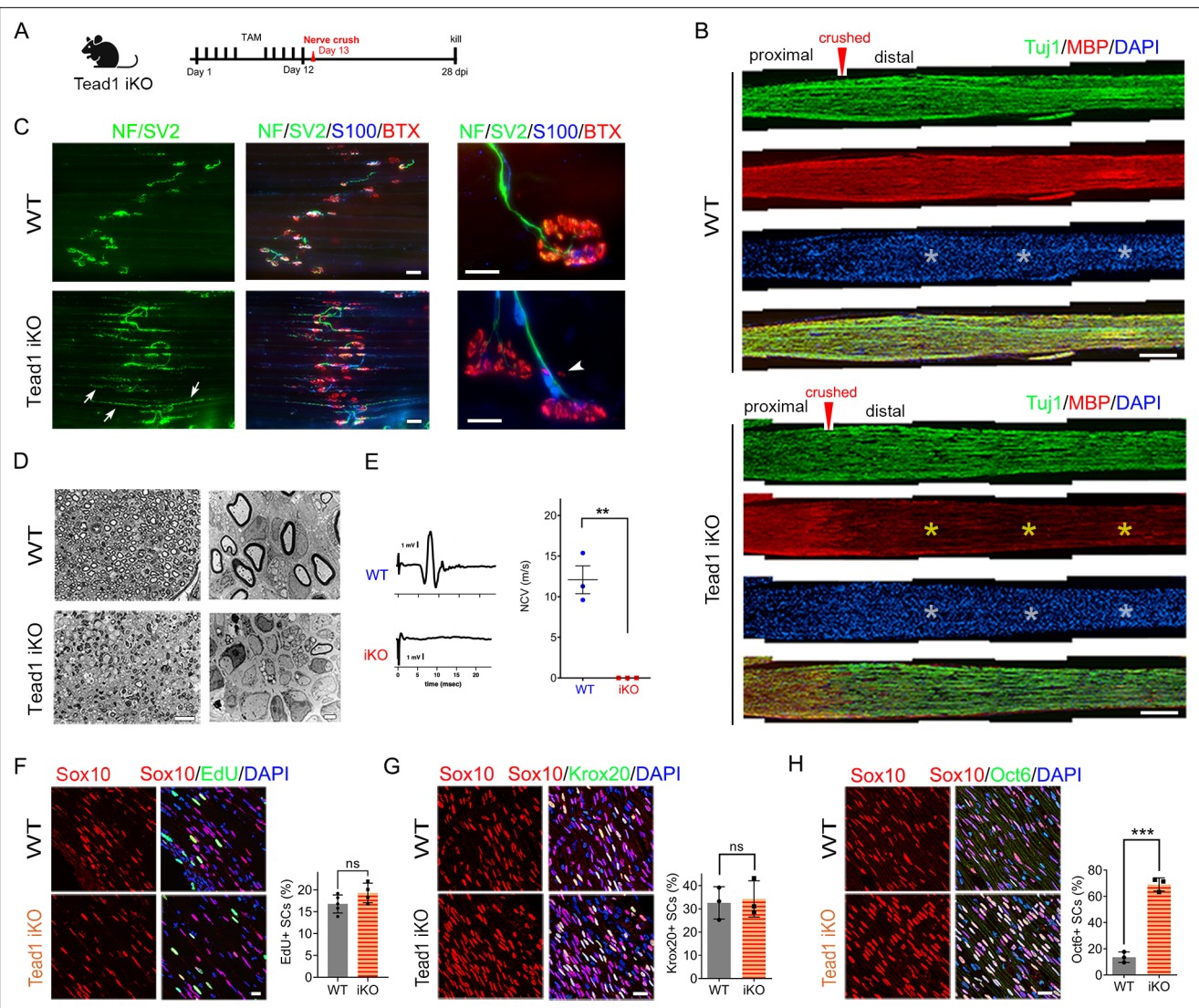

**Figure 8.** TEAD1 is required for functional regeneration of peripheral nerve. (**A**) Cartoon showing timeline of tamoxifen (TAM) injection and time of sacrifice of *Tead1* iKO mice after nerve crush. (**B**) Low magnification images of longitudinal sections of ~4 mm long SN segments of WT and *Tead1* iKO mice 28 days after nerve crush. Arrowheads denote crush site. Axons and myelin are marked by Tuj1 and MBP, respectively. All cell nuclei are marked by DAPI. Yellow asterisks denote the distal nerve segment of *Tead1* iKO, which almost completely lacks myelin. White asterisks denote distal segments of both WT and *Tead1* iKO, which exhibit more cells than proximal segments due to SC proliferation after injury. Scale bar: 500 µm (**C**) Representative low and high magnification NMJ images of WT and Tead1 iKO 28 days after nerve crush. Axon and axon terminals are marked by neurofilament (NF) and SV2 staining. SCs are marked by S100. Muscle acetylcholine receptors are marked by α-Bungarotoxin. Arrows denote examples of axon terminals that grew past NMJs in iKO. Arrowheads denote examples of extrasynaptic AChR clusters in iKO. Scale bar: 50 µm (low magnification panels), 20 µm (high magnification panels) (**D**) Representative semi-thin and TEM images of transverse sections of WT and *Tead1* iKO 28 days after nerve crush. Scale bar: 20 µm (left semi-thin panels), 2 µm (right TEM panels) (**E**) Representative CMAP recordings and NCV measurements of WT and *Tead1* iKO at 28 days after nerve crush. n=3 mice per genotype, **p≤0.01, Student's unpaired t-test. (**F**) Immunostaining and quantification of EdU +SCs on longitudinal sections of WT and *Tead1* iKO SNs at 28 days after nerve crush. SC nuclei are marked by Sox10. All cell nuclei are marked by DAPI. n=4–5 mice per genotype, ns, p>0.05, Student's unpaired t-test. Scale bar: 20 µm (**G**) Immunostaining and quantification of Krox20 +SCs on longitudinal sections of WT and *Tead1* iKO SNs at 28 days after nerve crush. SC nuclei are marked by Sox10. All cell nuclei are marked by DAPI. n=3 mice per genotype, ns, p>0.05, Student's unpaired t-test. Scale bar: 20 µm (**H**) Immunostaining and quantification of Oct6 +SCs on longitudinal sections of WT and *Tead1* iKO SNs at 28 days after nerve crush. SC nuclei are marked by Sox10. All cell nuclei are marked by DAPI. n=3 mice per genotype, ns, ***p<0.001, Student's unpaired t-test. Scale bar: 20 µm.

The online version of this article includes the following source data for figure 8:

**Source data 1.** This zip archive contains source files for graphs in *Figure 8E, F, G and H*.

## Unique, redundant, and opposing roles of TEAD1

Although SCs express other TEADs, *Tead1* cKO mice grossly phenocopy *Yap1/Wwtr1* cDKO mice. They exhibit abnormal SC proliferation, delayed radial sorting, blocked myelin formation and remyelination, and normal axon regeneration. Furthermore, *Tead1* cKO mice, like *Yap1/Wwtr1* cDKO, are defective in regulating Oct6, Krox20, myelin proteins, and enzymes that synthesize cholesterol. These similarities show that YAP/TAZ regulate both SC proliferation and differentiation, the two opposing processes crucial for myelination, by partnering mainly with TEAD1 for developmental myelination and remyelination after injury. We also observed multiple differences between *Tead1* cKO and *Yap1/Wwtr1* cDKO mice. Unlike *Yap1/Wwtr1* cDKO, *Tead1* cKO initiates myelin formation in some developing axons and maintains myelination in almost all adult axons. Additionally, in *Tead1* cKO, Krox20 is upregulated and a large excess of SCs is generated. It is also notable that, whereas some developing axons are myelinated in *Tead1* cKO, almost all axons regenerating after injury are unmyelinated. These differences are likely due to stage-dependent redundancy and opposing roles of TEADs. Consistent with this notion, loss of TEAD1 does not affect nuclear localization of YAP/TAZ, which depends on interactions with TEADs (*Diepenbruck et al., 2014*; *Kim et al., 2020*; *Kofler et al., 2018*; *Yao et al., 2018*). TEAD1-4 are all co-expressed in SCs through multiple stages of nerve development, maintenance, and regeneration (*Fernando et al., 2016*; *Grove et al., 2017*; *He et al., 2018*). Notably, nuclear retention of YAP/TAZ occurs in *Tead1* cKO and iKO SCs when TEAD1 has a unique role in SCs for developmental myelination and remyelination. Thus, other TEADs can retain nuclear YAP/TAZ, thereby supporting transcriptional activity of YAP/TAZ, but cannot substitute for some or most of the functions of TEAD1 in a stage-dependent manner.

It is also notable that, like YAP/TAZ, the unique roles of TEAD1 change dynamically in successive steps of myelination, even though other TEADs are present. Thus, TEAD1 is uniquely required for timely proliferation of SCs; uniquely opposes SC proliferation to promote cell-cycle exit after radial sorting; redundantly promotes upregulation of Krox20; and then uniquely promotes upregulation of myelin proteins and cholesterol biosynthetic enzymes. Such unique and redundant roles for TEAD1 in the presence of other TEADs are supported by cell-based transcription assays, in which YAP-TEAD4 can substitute for YAP-TEAD1 in upregulating transcription from the Krox20 promoter, but not from the MPZ intronic enhancer (*Fernando et al., 2016*). Stage-dependent function of TEADs was also reported in cortical development, another multi-step differentiation program. As in nerve development, multiple TEADs (TEAD1-3) are co-expressed throughout cortical development. Depending on the differentiation stage, TEAD1 uniquely promotes proliferation, redundantly promotes proliferation with TEAD2 and 3, or redundantly promotes differentiation with TEAD3, when it is opposed by TEAD2 (*Mukhtar et al., 2020*).

As *Yap1/Wwtr1* cDKO SCs arrest at the promyelination stage, we were unable to investigate the role of YAP/TAZ in myelin growth (*Grove et al., 2017*). In contrast, some *Tead1* cKO SCs can myelinate but fail to grow myelin sheaths proportional in size to axon diameter. Importantly, such myelinating SCs apparently lack TEAD1 because TEAD1 deletion is highly efficient in *Tead1* cKO mice. Support for a role of TEADs in the regulation of myelin sheath thickness comes from a study of TEAD4 cKO mice, in which SCs lacking TEAD4 gradually upregulate myelin protein expression, resulting in adult mice with abnormally thick myelin (*He et al., 2018*). Thus, it is possible that TEADs 1 and 4 oppose each other in the regulation of myelin sheath growth. Likewise, they may oppose each other in the regulation of SC proliferation during development, resulting in a large excess of SCs at P22 and P50 in *Tead1* cKO.

## TEAD1 regulation of Krox20 and myelin gene expression

Previous work suggests that YAP/TAZ-TEAD1 promote myelination in part through direct upregulation of Krox20 (*Deng et al., 2017*; *Grove et al., 2017*). Unlike *Yap1/Wwtr1* cKO mice, *Tead1* cKO mice upregulate Egr2/Krox20, but they fail to upregulate myelin proteins and cholesterol biosynthetic enzymes. Our findings extend previous work to suggest that TEAD1 functions redundantly in direct upregulation of Krox20, but that its function is required for upregulated Krox20 to promote myelination. Krox20 activity may be repressed in *Tead1* cKO by Oct6, which remains abnormally elevated. In support of this possibility, persistently elevated Oct6 overrides Krox20 to inhibit myelin gene expression (*Monuki et al., 1993*; *Ryu et al., 2007*). Other mice with defective lipid biosynthesis and a phenotype like *Tead1* cKO also have abnormally elevated Oct6 after Krox20 upregulation (*Norrmén*

*et al., 2014*; *Sherman et al., 2012*). An alternative possibility is that a transcriptional co-activator that synergizes with Krox20 is absent in *Tead1* cKO SCs. We speculate that TEAD1 itself may function as such a factor. Supporting this idea are reports that gene regulatory elements function by co-binding several TFs, which synergistically activate transcription (*Heinz et al., 2015*; *Ren and Yue, 2015*), as at the Krox20 MSE (Myelinating Schwann cell element) which is the enhancer responsible for Krox20 expression (*Ghislain and Charnay, 2006*; *Grove et al., 2017*; *He et al., 2010*; *Kao et al., 2009*). Krox20 synergizes with Sox10 to activate myelin genes (*Jones et al., 2007*) and with SREBP to activate lipid biosynthesis genes (*Leblanc et al., 2005*). Like Krox20, TEAD1 can directly promote transcription from PMP22 and MPZ myelin gene enhancers (*Fernando et al., 2016*; *Lopez-Anido et al., 2016*). Alternatively, or additionally, TEAD1 may function indirectly by upregulating co-factors, such as SREBP2, which is reduced in *Tead1* cKO. Notably, lipid biosynthesis is necessary to downregulate Oct6 and upregulate myelin gene transcription during developmental myelination (*Saher et al., 2009*; *Verheijen et al., 2009*). An interesting possibility, therefore, is that YAP/TAZ-TEAD1 and Krox20 may cooperatively downregulate Oct6 and upregulate myelin gene expression in part by promoting lipid biosynthesis.

## YAP/TAZ-TEAD1 regulation of cholesterol biosynthesis

SREBP1 levels are not altered in *Tead1* cKO or *Yap1/Wwtr1* cDKO, suggesting that fatty acid synthesis is not directly regulated by TEAD1 or YAP/TAZ. It is likely that the reduced levels of SCD1 at P8 and slight increase at P40 in *Tead1* cKO, and the large increase in SCD1 in *Yap1/Wwtr1* cDKO at P60 are due to developmental delay in these mutants. We did not observe downregulation of HMGCR in *Tead1* cKO or *Yap1/Wwtr1* cDKO mice at P8 or P40, which contradicts the report of reduced mRNA expression of HMGCR in *Yap1/Wwtr1* mutants at P3 and P5 (*Deng et al., 2017*; *Poitelon et al., 2016*); we suggest this difference is due to developmental delay in *Tead1* cKO and *Yap1/Wwtr1* cDKO. However, marked downregulation of IDI1 and FDPS in *Tead1* cKO, *Yap1/Wwtr1* cDKO, and *Yap1/Wwtr1* iDKO suggests that cholesterol biosynthesis is defective in these mutants. In support of this notion, SC-selective deletion of SCAP (SREBP cleavage activation protein), which is required to produce active SREBP1 and SREBP2 (*Verheijen et al., 2009*), or of Squalene synthase, an essential enzyme in cholesterol biosynthesis (*Saher et al., 2009*), evokes phenotypes which are remarkably similar to those of *Tead1* cKO mice: both mutants have an increased number of SCs, a severe delay at the promyelination stage, and persistently thin myelin. In addition, in SCAP cKO, Oct6 is upregulated continuously and Squalene synthase cKO fails to upregulate myelin gene transcription, resulting in severely reduced MBP, MPZ, and PMP22 protein levels.

How might YAP/TAZ-TEAD1 regulate IDI1 and FDPS? Multiple studies show that YAP/TAZ can activate the mTORC1 pathway (*Hu et al., 2017*; *Kim et al., 2015*), and that IDI1 and FDPS levels are reduced in SNs of Raptor cKO, a core component of mTORC1 (*Norrmén et al., 2014*). Furthermore, the Raptor cKO phenotype is like that of *Tead1* cKO, exhibiting an increased number of SCs, normal upregulation of Krox20, persistent upregulation of Oct6, delay at the promyelination stage, and thin myelin. However, HMGCR protein levels are reduced, and aberrant upregulation of Oct6 is transient (*Norrmén et al., 2014*). After nerve injury, formation of repair SCs is delayed in Raptor mutants (*Norrmén et al., 2018*), unlike TEAD1 and YAP/TAZ mutants. These differences suggest that YAP/TAZ-TEAD1 are not directly upstream of mTORC1. Another possibility is that YAP/TAZ-TEAD1 may directly regulate transcription of IDI1 and FDPS genes. Such direct regulation of cholesterol biosynthetic enzymes has been observed for other TFs in SCs (*Kim et al., 2018*; *Leblanc et al., 2005*).

## TEAD1 regulation of remak bundles

In both *Tead1* cKO and iKO mice, there is defective ensheathment of C-fibers by non-myelinating SCs in Remak bundles. This phenotype is distinct from those mutants in which SCs surround and occasionally myelinate bundles filled with large and small caliber axons, which are considered radial sorting defects (*Feltri et al., 2016*). Remak bundles are occasionally myelinated in WT SNs during early postnatal development, but this defect is corrected and does not persist in adults (*Rasi et al., 2010*). Thus, the defective Remak bundles that we observed in *Tead1* cKO are not due to developmental delay.

Remak bundles form normally at P60 in *Yap1/Wwtr1* cDKO, although many large bundles containing mixed caliber axons are present during development (*Grove et al., 2017*). The difference between *Tead1* and *Yap1/Wwtr1* mutants is supported by the presence of TEAD1 and absence of YAP/TAZ

in non-myelinating SC nuclei (*Grove et al., 2017*). Notably, defective SC ensheathment of C-fibers is observed in various SC developmental mutants, including those with defective SC basal lamina (*Rasi et al., 2010*) and overexpression of NRG1-IIIb (Neuregulin 1 type-IIIb) in axons (*Gomez-Sanchez et al., 2009*). Therefore, it is possible that TEAD1 mediates extracellular signaling from the SC basal lamina and/or axons that is responsible for forming and maintaining proper association of SCs with C-fibers. Notably, SC ensheathment of C-fibers is required for normal NCV (*Taveggia et al., 2005*), and NCV in the mutants overexpressing NRG1-IIIb is significantly reduced compared to WT (*Gomez-Sanchez et al., 2009*). Future studies will address NRG1 signaling and NCV in cKO and iKO mice lacking TEAD1 and/or other TEADs.

# Materials and methods

## Key resources table

| Reagent type (species) or resource | Designation | Source or reference | Identifiers | Additional information |
|---|---|---|---|---|
| Strain, strain background (*Mus musculus*) | C57Bl/6 | Jackson Laboratory | Stock #: 000664; RRID:IMSR JAX:000664 | |
| Genetic reagent (*M. musculus*) | Plp1-Cre-ERT2 | *Leone et al., 2003* | MGI:2663093 | |
| Antibody | anti-Yap/Taz (rabbit monoclonal) | Cell Signaling Technology | D24E4, #8418 RRID:AB_10950494 | IHC 1:200 Western 1:1000 |
| Antibody | anti-SCG10 (rabbit monoclonal) | Novus Biologicals | NBP1-49461 RRID:AB_10011569 | IHC 1:5000 |
| Antibody | anti-Yap (rabbit monoclonal) | Cell Signaling Technology | D8H1X, #14074 RRID:AB_2650491 | IHC 1:200 |
| Antibody | anti-Sox10 (goat polyclonal) | R&D Systems | #AF-2864 RRID:AB_442208 | IHC 1:100 |
| Antibody | anti-Sox10 (rabbit monoclonal) | Abcam | EPR4007, #ab155279 RRID:AB_2650603 | IHC 1:250 |
| Antibody | anti-Egr2 (rabbit polyclonal) | Professor Dies Meijer, University of Edinburgh | | IHC 1:4000 |
| Antibody | anti-Oct6 (rabbit monoclonal) | Abcam | EP5421, #ab126746 RRID:AB_11130256 | WB 1:1000 |
| Antibody | Anti-Oct6 (rabbit polyclonal) | Abcam | #ab31766 RRID:AB_776899 | IHC 1:800 |
| Antibody | Anti-c-Jun (mouse monoclonal) | BD Transduction Laboratories | #610326 RRID:AB_397716 | IHC 1:500 |
| Antibody | Anti-c-Jun (rabbit monoclonal) | Cell Signaling Technology | 60 A8, #9165 RRID:AB_2130165 | WB 1:1000 |
| Antibody | Anti-pS63-c-Jun (rabbit polyclonal) | Cell Signaling Technology | #9261 RRID:AB_2130162 | IHC 1:100 |
| Antibody | anti-Ki67 (rabbit polyclonal) | Abcam | #ab15580 RRID:AB_443209 | IHC 1:200 |
| Antibody | anti-p75NGFR (goat polyclonal) | Neuromics | #GT15057 RRID:AB_2737189 | IHC 1:400 |
| Antibody | anti-Tubulin β3 (rabbit polyclonal) | Biolegend | #802001 RRID:AB_2564645 | IHC 1:1000 |
| Antibody | IRDye-680 (goat anti-mouse) | LI-COR | #926–32220 RRID:AB_621840 | WB 1:15,000 |
| Antibody | HRP-Goat anti-mouse secondary antibody | Jackson Immunoresearch | #715-035-150 RRID:AB_2340770 | WB 1:12,000 |
| Antibody | HRP-Goat anti-rabbit secondary antibody | Jackson Immunoresearch | #115-055-062 RRID:AB_2338533 | WB 1:12,000 |
| Chemical compound, drug | Araldite 6005 | EMS | #10920 | |
| Chemical compound, drug | DDSA | EMS | #13710 | |
| Chemical compound, drug | DBP | EMS | #13101 | |

*Continued on next page*

*Continued*

| Reagent type (species) or resource | Designation | Source or reference | Identifiers | Additional information |
|---|---|---|---|---|
| Chemical compound, drug | BDMA | EMS | #11400–25 | |
| Other | Coated grids (100 mesh) | EMS | #FF100-Cu | |
| Chemical compound, drug | Osmium tetroxide (4% solution) | EMS | #19170 | |
| Chemical compound, drug | Lead nitrate | EMS | #17900 | |
| Chemical compound, drug | Sodium citrate | EMS | #21140 | |
| Chemical compound, drug | Uranyl acetate | EMS | #22400 | |
| Chemical compound, drug | Sodium borate | EMS | #21130 | |
| Chemical compound, drug | Toluidine blue | EMS | #22050 | |
| Chemical compound, drug | Paraformaldehyde | Sigma-Aldrich | #158127 | |
| Commercial assay or kit | Click-It EdU Alexa Fluor 594 kit | ThermoFisher Scientific | #C10339 | |
| Chemical compound, drug | EdU | ThermoFisher Scientific | #E10187 | |
| Chemical compound, drug | Tamoxifen | Sigma-Aldrich | #T5648 | |
| Other | DAPI stain | Invitrogen | #D1306 | IHC 1:250 |
| Antibody | Alexa 488, 568 or 647 secondaries | Jackson Immunoresearch | | IHC 1:250 to 1:1000 |
| Software, algorithm | Image Studio Lite | LI-COR, Inc | | |
| Software, algorithm | Prism | GraphPad Software, Inc | | |
| Software, algorithm | Stata | StataCorp LP | | Mann-Whitney test |

## Animals

All surgical procedures and animal maintenance complied with the National Institute of Health guidelines regarding the care and use of experimental animals and were approved by the Institutional Animal Care and Use Committee (Protocol# 4920) of Temple University, Philadelphia, PA, USA. Generation and genotyping of mice carrying targeted loxP sites in the *Tead1* gene have been described previously (*Wen et al., 2017*). Targeted ablation of *Tead1* in *Schwann cells* of embryonic mice was achieved by crossing *Tead1*$^{fl/fl}$ mice with mice transgenic for *Cre* driven by the *Mpz* promoter (*Tead1* cKO mice). Targeted ablation of *Tead1* in *Schwann cells* of adult mice was by crossing *Tead1*$^{fl/fl}$ mice with mice transgenic for the inducible Cre recombinase *Cre-ERT2* under control of the *Plp1* promoter (*Tead1* iKO mice). Control mice contained floxed *Tead1* alleles but were not transgenic for *Cre* or *Cre-ERT2. Mpz-cre;Yap1*$^{fl/fl}$; *Wwtr1*$^{fl/fl}$ (*Yap1/Wwtr1* cDKO) and *Plp1-creERT2; Yap1*$^{fl/fl}$; *Wwtr1*$^{fl/fl}$ (*Yap1/ Wwtr1* iDKO) mice used in this study were generated and genotyped as described previously (*Grove et al., 2017*). Both male and female mice were used in all experiments and were maintained on the C57BL/6 background.

## Tamoxifen administration

Tamoxifen was prepared and injected into 6- to 8-week-old *Tead1* or *Yap1/Wwtr1* mutant mice as previously described (*Grove et al., 2017*). A 10 mg/ml solution of tamoxifen was made in 10:1 sunflower oil: 100% ethanol. This solution was injected intraperoneally at a concentration of 0.2 mg/g body weight. Injection was once daily for 5 days, followed by a 2-day break, then once daily for 5 consecutive days.

## Nerve crush

Sciatic nerves of right hindlimbs were crushed 24 hr after the final tamoxifen injection, as described previously (*Grove et al., 2020*; *Son and Thompson, 1995*). A small skin incision was made in the posterior thigh and calf of the animals anesthetized by isoflurane. The sciatic nerve was crushed with a fine forceps (#5) for 10 s (3 X) adjacent to the sciatic notch. The crush site was marked using

charcoal-coated forceps, and the wound was closed. To identify proliferating *Schwann cells,* we intraperitoneally injected EdU (80 µg/g) eighty minutes before euthanasia, as previously described (*Grove et al., 2017*).

## Western blotting

After removal of epineurium and perineurium, sciatic nerves were snap-frozen in liquid nitrogen and stored at –80 °C. For Western blotting, sciatic nerves were lysed in RIPA buffer (25 mM Tris pH 7.5, 150 mM NaCl, 1% Triton X-100, 0.5% sodium deoxycholate, 1 mM EDTA, 0.1% SDS); for immunoprecipitation, lysis was in co-IP buffer (50 mM Tris pH 8.0, 150 mM NaCl, 0.5% Triton X-100, 10% glycerol, 1 mM EDTA), both containing protease and phosphatase inhibitors. Lysis was for 40 min on ice, followed by microcentrifugation at 14,000 rpm for 20 min. Protein concentration was determined using the BCA assay (ThermoFisher).

Primary antibodies were used at the following concentrations for Western blots: rabbit anti-Krox20 (Covance; 1:400), rabbit anti-Oct6 (Abcam ab126746; 1:1000), mouse anti-Myelin Basic Protein (Covance SMI94; 1:1000), mouse anti-beta actin (SIGMA-Aldrich A5441; 1:5000), rabbit anti-TEAD1 (Cell Signaling D9X2 L, #12292; 1:1000), rabbit anti-TEAD2 (Novus Biologicals, #NBP3-05267; 1:1000), rabbit anti-TEAD3 (Cell Signaling #13224; 1:1000), rabbit anti-TEAD4 (Santa Cruz, #sc-101184; 1:1000), rabbit anti-pan-TEAD (Cell Signaling, #13295; 1:1000), rabbit anti-YAP/TAZ (Cell Signaling D24E4, #8418; 1:1000, 1:100 for immunoprecipitations), rabbit anti-phospho-YAP(Ser127) (Cell Signaling D9W2I #13008; 1:1000), rabbit anti-phospho-YAP(Ser397) (Cell Signaling D1E7Y #13619; 1:1000), rabbit-anti-MPZ (Cell Signaling #57518; 1:1000), mouse anti-SREBP1 (Novus Biologicals, #NB600-582; 1:1000), rabbit anti-SCD1 (Cell Signaling, #2794; 1:1000), rabbit anti-SREBP2 (Bethyl laboratories, #A303-125A), rabbit anti-HMGCR (Novus Biologicals, #NBP2-66888; 1:1000), rabbit anti-FDPS (Proteintech, #16129–1-AP; 1:1000), rabbit anti-IDI1 (Proteintech, #11166–2-AP; 1:1000). Control whole rabbit IgG for immunoprecipitations was from Jackson Laboratories (011-000-003) and was used at 1 µg/ml, which was several-fold greater than all other antibodies used for immunoprecipitations. Secondary antibodies for Western blotting were HRP-linked sheep anti-mouse IgG and donkey anti-rabbit (both GE Healthcare; 1:5000) and HRP-linked mouse anti-rabbit, light-chain specific (Jackson ImmunoResearch 211-032-171; 1:20,000). Blots were developed using ECL, ECL Plus (GE Healthcare) or Odyssey (LiCor).

## Immunohistochemistry, confocal and widefield fluorescence microscopy

For immunostaining, sciatic nerves were removed and immediately fixed in 4% paraformaldehyde in PBS for 1 hr on ice. Nerves were washed 3 times in PBS, then stored in 15% sucrose in PBS overnight at 4 °C for cryoprotection. Nerves were frozen-embedded in cryoprotectant medium (Thermo Fisher Scientific, Waltham, MA) in isomethylbutane at –80 °C. Seven to 10 µm sections from the nerves were cut using a cryostat (Leica Microsystems, Germany) and collected directly onto glass slides. For immunolabeling, nerve sections were rehydrated in PBS, permeabilized in 0.5% Triton/PBS for 20 min, washed with PBS, then blocked in 2% bovine serum albumin (BSA) in PBS for 1 hr. Sections were incubated with primary antibodies in blocking buffer overnight at 4 °C in a hydrated chamber, washed with PBS, and incubated with secondary antibodies in blocking buffer for 2 hr at room temperature. Sections were washed with PBS, stained with DAPI for 10 min, and mounted with Vectashield mounting medium (Vector Labs, Burlingame, CA).

Primary antibodies were used at the following concentrations for immunostaining of cryosections: guinea pig anti-Sox10 (kind gift from Michael Wegner, University of Erlangen, Bavaria, Germany; 1:1000), rabbit anti-Krox20 (Covance; 1:400), rabbit anti-Oct6 (Abcam ab31766; 1:800), rabbit anti-Ki67 (Novocastra NCL-Ki67p; 1:200), rabbit anti-Neurofilament 200 (SIGMA-Aldrich N4142; 1:500), mouse anti-Neurofilament (Covance SMI 312; 1:1000), mouse anti-Tubulin β3 (clone Tuj1, Covance #MMS-435P; 1:1000), mouse anti-SV2 (Developmental Studies Hybridoma Bank, Iowa; 1:10), rabbit anti-S100 (DAKO; 1:600), mouse anti-Myelin Basic Protein (Covance SMI 94; 1:2000), rabbit anti-TEAD1 (Cell Signaling D9X2 L, #12292; 1:200), rabbit anti-pan-TEAD (Cell Signaling D3F7L, #13295; 1:200), rabbit anti-YAP/TAZ (Cell Signaling D24E4, #8418; 1:200).

For immunostaining of extensor digitorum longus (EDL) and soleus muscles, mice were perfused with 4% paraformaldehyde, and the muscles removed and post-fixed for 20 min at room temperature. Muscles were then incubated for 15 min with rhodamine-conjugated α-bungarotoxin (Invitrogen,

Carlsbad, CA), permeabilized in −20 °C methanol for 5 min, and blocked for 1 hr. in PBS containing 0.2% Triton and 2% BSA. The muscles were subsequently incubated overnight at 4 °C in a cocktail of primary antibodies diluted in the blocking solution. Axons and nerve terminals were labeled with mouse monoclonal antibodies to neurofilaments (SMI 312; Sternberger Monoclonals, Baltimore, MD) and to a synaptic vesicle protein, SV2 (Developmental Studies Hybridoma Bank, Iowa City, IA). *Schwann cells* were labeled with rabbit anti-cow S-100 polyclonal antibody (Dako, Carpentaria, CA). After incubation with the primary antibodies, muscles were rinsed in PBS and incubated with secondary antibodies in the blocking solution, for 1 hr at room temperature. The muscles were then rinsed in PBS and prepped for whole-mount by peeling the top layer of muscle fibers using microscissors and forceps, as previously described (*Son and Thompson, 1995*). The resulting 'muscle fillets' were mounted on a slide and coverslipped.

Conventional fluorescence microscopy was performed using Olympus BX53 and Zeiss Axio Imager 2 microscopes. Images were captured using Hamamatsu ORCA R2 C10600 or Zeiss Axiocam HRm Rev 3 digital cameras, respectively, and Metamorph software. For confocal microscopy, we used a Leica SP8 confocal microscope, with either a 40x1.3 NA or 63x1.4 NA objective and Leica proprietary software. Acquired stacks were assembled using the maximum projection tool, and figures were prepared using Adobe Photoshop.

## Electron microscopy, histology, and morphometry

Mice were perfused intravascularly with 2.5% glutaraldehyde and 4% paraformaldehyde in 0.1 M cacodylate buffer pH 7.4. Sciatic nerves were removed and incubated in the same buffer for 90 min at room temperature, then overnight at 4 °C. Specimens were post-fixed for 1 hr with $OsO_4$, dehydrated, then embedded in araldite. Semi-thin (0.5 μm) and ultrathin (70 nm) sections were cut using an ultramicrotome and stained with toluidine blue or uranyl acetate and lead citrate, respectively. Semi-thin sections were examined using bright field on an Olympus BX53 microscope, and images were captured using Hamamatsu ORCA R2 C10600 digital camera and Metamorph software. For analysis of the number of myelin profiles in control vs mutant sciatic nerves, semi-thin sections from proximal-, mid- and distal regions of the sciatic nerve were taken, and 2 non- overlapping images of each section were obtained using the 60 x objective. For electron microscopy, stained ultrathin sections were examined using a JEOL 1010 electron microscope fitted with a Hamamatsu digital camera and AMT Advantage image capture software.

## Proliferation assays

The percentage of *Schwann cells* in S-phase was measured by EdU incorporation. Mice were injected intraperitoneally and subcutaneously with 80 μg/ g body weight of EdU dissolved in PBS. Eighty min. later, mice were sacrificed, and sciatic nerves removed and fixed for 1 hr in 4% paraformaldehyde; longitudinal 7 μm cryosections were used for analysis. EdU incorporation was identified using the Click-iT EdU Alexa Fluor 555 labeling kit (Molecular Probes); the recommended protocol was followed, and labeling was for 1 hr. After EdU labeling, sections were immunostained with anti-Sox10 to identify *Schwann cells*. The percentage of *Schwann cells* (Sox10 +nuclei) that were EdU +was determined by counting confocal images using ImageJ software.

## Electrophysiology

The procedure for measuring nerve conduction velocity was an adaption of a previously used protocol (*Schulz et al., 2014*). Mice were deeply anesthetized, and hair was completely removed from the hindquarters. A 27 g percutaneous needle electrode was inserted subcutaneously over the femur to stimulate the sciatic nerve at two locations: at the hip joint and 1 cm distal to this site. The stimulation reference electrode (26 G percutaneous needle) was inserted over the back. Bipolar EMG electrodes were inserted subcutaneously in the digital interosseous muscles to record compound action potentials in response to activation of the sciatic nerve. Stimulation and recording were conducted on both hindlimbs. The muscle response was first tested with single pulses (100 μs duration) to determine the threshold and level of activation producing a stable compound action potential in the muscle. The level of stimulation which produced stable responses was then used for two series of stimulations with 2–20 s long trains of biphasic square wave pulses (100 μs duration) delivered at 1–5 Hz. The nerve was rested for 2 min between runs. Stimulation was delivered via an isolated pulse stimulator (A-M

Systems Model 2100, A-M Systems Inc, Carlsborg, WA). EMG compound muscle action potential (CMAP) responses were recorded with a differential AC amplifier (A-M Systems Model 1700; gain 1 k-10k; pass band 10–50 KHz). Data was sampled at 50 KHz using LabVIEW (National Instruments Corporation, Austin, TX) running on a personal computer. Multiple recordings were taken at each location to ensure a minimum of 10 representative responses for each location and animal. Electro-physiological data were analyzed using Igor Pro (Wavemetrics Inc, Lake Oswego, OR).

## Data analysis

In each experiment, data collection and analysis were performed identically, regardless of mouse genotype. We used the sample (replicate) size (number) typically used in earlier studies of peripheral myelination. Data are presented as mean +/-SD. Statistical analysis was done using Student's unpaired t-test for two-group comparisons and analysis of variance (ANOVA) with Tukey's test for multiple comparisons, according to the number of samples and the analysis of mice at multiple ages. Normal distribution of all data was tested using Shapiro-Wilk normality test. Sample sizes were similar to those conventional in the field and are indicated in the main text, methods, or figure legends. A p-value of 0.05 or less was considered statistically significant.

## Acknowledgements

We thank Alan Tessler and members of the Son laboratory for critical reading of the manuscript. We thank Dr. Eric Olson for *Yap1* and *Wwtr1* floxed mice, Drs. Ueli Suter and Kelly Monk for *Plp1*-creERT2 mice. *Plp1*-creERT2 mice were generated by Dr. Ueli Suter using a patented Cre-ERT2 construct developed by Dr. Pierre Chambon at GIE-CERBM. This work was supported by grants from Shriners Children's (#84050 to Y-J.S.) and NIH NINDS (NS105796 to Y-J.S.).

## Additional information

### Funding

| Funder | Grant reference number | Author |
|---|---|---|
| National Institute of Neurological Disorders and Stroke | NS105796 | Young-Jin Son |
| Shriners Hospitals for Children | 84050 | Young-Jin Son |

The funders had no role in study design, data collection and interpretation, or the decision to submit the work for publication.

### Author contributions

Matthew Grove, Formal analysis, Investigation, Writing – original draft; Hyukmin Kim, Michel Lemay, Formal analysis, Investigation; Shuhuan Pang, Investigation, Methodology; Jose Paz Amaya, Formal analysis; Guoqing Hu, Jiliang Zhou, Resources, Methodology; Young-Jin Son, Conceptualization, Supervision, Funding acquisition, Investigation, Writing – review and editing

### Author ORCIDs

Hyukmin Kim http://orcid.org/0000-0002-3270-4681
Michel Lemay https://orcid.org/0000-0002-5636-0297
Young-Jin Son http://orcid.org/0000-0001-5725-9775

### Ethics

All surgical procedures and animal maintenance complied with the National Institute of Health guide-lines regarding the care and use of experimental animals and were approved by the Institutional Animal Care and Use Committee (Protocol# 4920) of Temple University, Philadelphia, PA, USA.

### Decision letter and Author response

Decision letter https://doi.org/10.7554/eLife.87394.sa1

Author response https://doi.org/10.7554/eLife.87394.sa2

## Additional files

### Supplementary files
- MDAR checklist

### Data availability
All data generated or analysed during this study are included in the manuscript and supporting files; Source Data files have been provided for all the figures.

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
