## [Editor Report]

This important study demonstrates that the transcription factor TEAD1 is required for the function of Yap/Taz in Schwann cells, with conditional mouse mutants having very similar dysmyelinated phenotypes. Convincing histological evidence is shown for the role of TEAD1 itself, leaving open the function of other TEAD proteins in this system. This study will nevertheless be of great interest to researchers in the field of peripheral nerve development.

---

## [Decision Letter]

**Decision letter after peer review:**

Thank you for submitting your article "TEAD1 is crucial for myelination, Remak bundles and functional regeneration of peripheral nerves" for consideration by *eLife*. Your article has been reviewed by 3 peer reviewers, and the evaluation has been overseen by a Reviewing Editor and Anna Akhmanova as the Senior Editor.

The reviewers have discussed their reviews with one another, and the Reviewing Editor.

As you will also see below the reviewers found your study interesting and suitable as a research advance over your previous work on YAP/TAZ in the peripheral nervous system. However, it was felt strongly that more information is needed about the role of other TEAD family members in Schwann cell development and myelination, which we hope is possible in a revised version. The referees also suggest that some restructuring of the manuscript would improve its readability. Please provide a point-to-point response to explain your revised manuscript.

With the results of this revision work at hand, the editor will also draft a summarizing assessment to be posted alongside the preprint.

*Reviewer #1 (Recommendations for the authors):*

1. Despite the extensive analyses I have found it quite difficult to follow the outflow of the manuscript with the continuous parallelism with their previous studies and the comparison with the results on YAP/TAZ dKO mice. If the readers are not familiar with the specific topic they will not be able to follow the argument.

2. The conclusions reached in the IP experiment shown in Figure 2 are too speculative. The quality of the western blot is poor and it is not possible to infer that "TEAD1 is greatly favored as the binding partner for YAP/TAZ" by simply looking at how the bands obtained in this analysis run on the gel.

3. The proposed correlation between OCT6 and Krox 20 in this model is intriguing but not supported by the experimental data, which simply rely on the expression levels of these proteins.

4. Since TEAD1 has basically no role in myelin maintenance the conclusions of this part should be softened, unless corroborated by a more precise analysis on the number of a-myelinated axons and the extent of the mild demyelination. Is the latter accompanied by axonal degeneration?

5. Sometimes the writing is not clear and it is difficult to follow, as in the case of the conclusions relative to Figure 4.

*Reviewer #2 (Recommendations for the authors):*

The authors need to address the following major and minor points before publication can be granted:

1. Antibodies against murine Tead1-4 are used to discriminate against Tead1-4 proteins isolated from sciatic nerve cell lysates (Figure 1). While the specificity of the Tead1 antibody is convincing, this has not been shown for the other antibodies, including the pan-Tead antibody. To confirm the other antibodies' specificity, the authors need to e.g., use shRNA-treated (or CRISPRi or knockout) murine cells/tissue for Tead2, 3, and 4. For the pan-Tead antibody, this needs to be conducted in a combinatorial manner. The functionality of the antibodies is central to this manuscript, as the authors claim that Tead1 is the key player to interact with YAP1/TAZ in co-IPs. In the text, the authors state that "other TEADs can complex with YAP/TAZ" (line 242/243), but this has not been directly shown for TEAD2, 3, and 4. Therefore, the authors need to individually address these interactions as well and need to quantitatively show (e.g., in co-IPs) how Tead1, 2, 3, and 4 bind to YAP1/TAZ.

2. In Tead1 cKO, SCs increased twice as compared to SCs of WT, measured at P22 and P50. How can this be mechanistically explained? The authors describe that Tead1 is the main TF of the TEAD family to interact with Yap1/Taz in SCs. However, could it be that Tead2-4 have a compensatory role here? And what is the contribution of such a potentially compensated Tead in WT SCs? Therefore, further experiments (e.g., qPCR on Tead2-4) are needed to validate and clarify this.

3. Oct6 protein level is upregulated in Tead1 cKO SCs. As with Tead family proteins, Oct6 belongs to a protein family, coving multiple genes, like Oct-1, Oct-2, Oct-3/4. Are those regulated in SCs? If so, how? This is particularly important for Oct-4 as YAP1 can regulate Oct-4 transcription. Further, why do the authors see a clear upregulation of Krox20 in the Western blot analysis, but not in the ICC staining? For Oct6, however, the results nicely correlated in WB and ICC.

4. In general, the selected targets the authors analysed are reasonable. However, the study will benefit from a more comprehensive and unbiased approach by conducting a transcriptomic analysis in Tead1 cKO SC to identify the entire deregulated network. This can be done using RNA-seq on isolated SNs (or even single-cell RNA-seq, if required to differentiate among cell types). The authors may want to also include SNs from Yap/Taz dKO SNs to better understand the differences between Tead1 ko and the Yap/Taz dKO SNs. The ambiguity and the rather high degree of uncertainty for some mechanistic aspects are also reflected in the discussion where many options were considered but a clearly preferred mechanism is not provided.

5. The effect of an induced Tead1 inactivation in SC is tested using a tamoxifen-inducible PLP specific Cre recombinase (Figure 7). However, data on the induced inactivation of Tead1 is missing. To validate this Tead1 iKO model, the authors must provide data (e.g., WB). Also, what is the percentage of tamoxifen-induced Tead1 iKO SC? The authors provide clear evidence for an effect on Krox20 in Yap1/Taz iDKO SC after 2 weeks of tamoxifen treatment. In Tead1 iKO SC, however, there is no effect after 6 weeks of tamoxifen treatment. How is this after 2 weeks? Could this be seen in Western blot analyses (c.f. to Tead1 cKO)? Also, how is the response to Oct6+ SC after 2 weeks? How is the response in Western blot analyses on targets of the cholesterol pathway (-> 2 weeks vs. 6 weeks)?

6. Statistics section: Were all data tested for normal distribution (-> Shapiro Wilk test) before applying the indicated tests?

*Reviewer #3 (Recommendations for the authors):*

The data are well-presented and solid. However, I have three suggestions that the authors might wish to address.

1) I found the presentation of the inducible TeaD1 KO in Figure 1 very confusing. These mutants are analysed and presented in Figures 7 and 8. Why not introduce them there? The point of Figure 1 is mainly to demonstrate the effective deletion of TeaD1 from the Schwann cell lineage and its effect on the other TEAD proteins expressed in wild type and in the absence of TEAD1 (Figure1A and 1D).

2) The timelines in Figure 7 and 8 should include the age at which TAM injection is started to avoid confusion on the age of the mice.

3) The authors should provide a much more detailed description of their YAP/TAZ IP experiments. In Figure 2F they present Western blots of IP of YAP/TAZ that appear close to 100% efficient, which is kind of incredible but possible of course. But what is really puzzling is the complete absence of a rabbit IgG heavy chain signal on their blots. They should explain the absence of this signal.

---

## [Author Response]

Reviewer #1 (Recommendations for the authors):1. Despite the extensive analyses I have found it quite difficult to follow the outflow of the manuscript with the continuous parallelism with their previous studies and the comparison with the results on YAP/TAZ dKO mice. If the readers are not familiar with the specific topic they will not be able to follow the argument.

As suggested, we now minimize the parallelism by removing our references to earlier studies on Yap/Taz cDKO. We keep them only when readers, including those not familiar with YAP/TAZ-TEAD(s) signaling, need the information to evaluate our claims regarding TEAD1 in YAP/TAZ regulation of myelin development, maintenance, and remyelination.

2. The conclusions reached in the IP experiment shown in Figure 2 are too speculative. The quality of the western blot is poor and it is not possible to infer that "TEAD1 is greatly favored as the binding partner for YAP/TAZ" by simply looking at how the bands obtained in this analysis run on the gel.

In accordance with this assessment and *eLife*’s revision policy, we have removed the IP/Western blot data (Figure 2F) that we used to claim that YAP/TAZ favor TEAD1 but are complexed with other TEADs in the absence of TEAD1. We will obtain additional biochemical and genetic data when the necessary resources for studying TEAD2-4 are available.

3. The proposed correlation between OCT6 and Krox 20 in this model is intriguing but not supported by the experimental data, which simply rely on the expression levels of these proteins.

Persistent upregulation of Oct6 has been observed in various mutant mice defective in Schwann cell myelination and aberrant overexpression of Oct6 overrides Krox20 to inhibit upregulation of myelin proteins (Monuki, Kuhn, & Lemke, 1993; Ryu et al., 2007). It remains enigmatic, however, how high levels of Oct6 inhibit Krox20 or myelin gene expression.

It is indeed intriguing to find that Oct6 is upregulated in Tead1 cKO because it suggests the possibility that Oct6 may repress Krox20 to prevent myelination in Tead1 cKO. Alternatively, Oct 6 upregulation may be a consequence of defective YAP/TAZ-TEAD1 signaling and Tead1 cKO may fail to myelinate normally due to the lack of transcriptional co-activator that synergizes with Krox20. We indeed favor this latter possibility and speculate that TEAD1 itself may function as such a cofactor. We have clarified and elaborated our comments regarding Oct6 in the Discussion.

4. Since TEAD1 has basically no role in myelin maintenance the conclusions of this part should be softened, unless corroborated by a more precise analysis on the number of a-myelinated axons and the extent of the mild demyelination. Is the latter accompanied by axonal degeneration?

We have now modified the title by inserting ‘developmental’ to indicate that TEAD1 is indispensable explicitly for developmental myelination. The new tile is that “TEAD1 is crucial for developmental myelination, Remak bundles…”. We have also softened our statements about the role of TEAD1 in myelin maintenance. We also mention the low incidence of demyelinated axons which were observed on discrete regions of SNs, indicating local, segmental demyelination. We observed no degenerating axons on TEM analysis.

5. Sometimes the writing is not clear and it is difficult to follow, as in the case of the conclusions relative to Figure 4.

We appreciate the suggestion and have amended our writing throughout the manuscript, including the conclusions relative to Figure 4 (now Figure 3).

Reviewer #2 (Recommendations for the authors):The authors need to address the following major and minor points before publication can be granted:1. Antibodies against murine Tead1-4 are used to discriminate against Tead1-4 proteins isolated from sciatic nerve cell lysates (Figure 1). While the specificity of the Tead1 antibody is convincing, this has not been shown for the other antibodies, including the pan-Tead antibody. To confirm the other antibodies' specificity, the authors need to e.g., use shRNA-treated (or CRISPRi or knockout) murine cells/tissue for Tead2, 3, and 4. For the pan-Tead antibody, this needs to be conducted in a combinatorial manner. The functionality of the antibodies is central to this manuscript, as the authors claim that Tead1 is the key player to interact with YAP1/TAZ in co-IPs. In the text, the authors state that "other TEADs can complex with YAP/TAZ" (line 242/243), but this has not been directly shown for TEAD2, 3, and 4. Therefore, the authors need to individually address these interactions as well and need to quantitatively show (e.g., in co-IPs) how Tead1, 2, 3, and 4 bind to YAP1/TAZ.

We appreciate the reviewer’s suggestion. We were able to demonstrate the specificity of Tead1 antibody and its expression in Schwann cell nuclei primarily because floxed Tead1 allele and IHC-compatible antibodies are available for TEAD1. As mentioned earlier, unlike TEAD1, the key resources for studying TEAD2-4 are lacking and our in vivo approach using AAV-shRNAs was not successful. We have, therefore, removed the co-IP data (Figure 2F) and the associated claim that ‘YAP/TAZ favors TEAD1 but can complex with other TEADs in the absence of TEAD1’. We consider this claim to be a minor portion of the manuscript and hope that the reviewers will adhere to *eLife*’s revision policy, which advises ‘either limit claims to those supported by data in hand, or to explicitly state that the relevant conclusions require additional supporting data’.

2. In Tead1 cKO, SCs increased twice as compared to SCs of WT, measured at P22 and P50. How can this be mechanistically explained? The authors describe that Tead1 is the main TF of the TEAD family to interact with Yap1/Taz in SCs. However, could it be that Tead2-4 have a compensatory role here? And what is the contribution of such a potentially compensated Tead in WT SCs? Therefore, further experiments (e.g., qPCR on Tead2-4) are needed to validate and clarify this.

TEADs in Schwann cells can be contradictory/opposing as well as compensatory to each other, as we discussed. For example, as we stated in the Discussion, Schwann cells ‘lacking’ TEAD4 developed thicker myelin (He et al., 2018), whereas those lacking TEAD1 failed to do so, as we found in the present study. It is possible, therefore, that TEAD4 may oppose TEAD1 in the regulation of Schwann cell proliferation, resulting in gradual increase of Schwann cells measured at P22 and P50 in Tead1 cKO.

It is also noteworthy that single cell RNAseq data show that Schwann cells express all 4 Teads throughout development and homeostasis (Gerber et al., 2021). Therefore, studying compensatory and/or opposing roles of TEAD2-4 requires a large amount of functional analysis with aforementioned key resources that are currently lacking. Simple quantitative analysis will not be able to determine their compensatory and/or opposing roles. For example, qPCR cannot demonstrate Schwann cell-specific expression of Tead2-4 because non-Schwann cells are likely to express them, as they do TEAD1.

3. Oct6 protein level is upregulated in Tead1 cKO SCs. As with Tead family proteins, Oct6 belongs to a protein family, coving multiple genes, like Oct-1, Oct-2, Oct-3/4. Are those regulated in SCs? If so, how? This is particularly important for Oct-4 as YAP1 can regulate Oct-4 transcription. Further, why do the authors see a clear upregulation of Krox20 in the Western blot analysis, but not in the ICC staining? For Oct6, however, the results nicely correlated in WB and ICC.

Of 11 known Oct proteins, Oct1 and Oct6 have been reported in developing Schwann cells, and, Schwann cell mRNA transcripts data consistently suggest little or no expression of other Oct mRNAs including Oct4 (Pou5f1) (Gerber et al., 2021) (https://snat.ethz.ch/index.html). Furthermore, Oct1 and Oct6 show distinct expression patterns, presumably reflecting their unique function in Schwann cells (Blanchard et al., 1996; the Schwann cell transcripts data). It is also noteworthy that upregulated Oct6 may not be the cause for the failure of Krox20 to upregulate myelin proteins in Tead1 cKO, as we now elaborate in our discussion. Based on these data, we respectfully disagree with the reviewer that studying expression of other Oct proteins and their regulation by TEAD1 is necessary for the present work.

Based on WB analysis, Oct6 is increased ~20 times, whereas Krox20 is increased ~4 times higher in Tead1 cKO than WT (Figure 4A). As Tead1 cKO contains twice as many Schwann cells as WT, WB data therefore indicate that Tead1 cKO Schwann cells contain only two-fold more Krox20 than WT Schwann cells (but 10 fold more Oct6). We have tested several Krox20 antibodies in ICC, but none appear to have sufficient sensitivity to reveal the two-fold increase of Krox20. On the other hand, Oct6 is not present in WT Schwann cells, unlike Krox20. Therefore, demonstrating 10-fold more Oct6 in Tead1 cKO did not require a highly sensitive antibody.

4. In general, the selected targets the authors analysed are reasonable. However, the study will benefit from a more comprehensive and unbiased approach by conducting a transcriptomic analysis in Tead1 cKO SC to identify the entire deregulated network. This can be done using RNA-seq on isolated SNs (or even single-cell RNA-seq, if required to differentiate among cell types). The authors may want to also include SNs from Yap/Taz dKO SNs to better understand the differences between Tead1 ko and the Yap/Taz dKO SNs. The ambiguity and the rather high degree of uncertainty for some mechanistic aspects are also reflected in the discussion where many options were considered but a clearly preferred mechanism is not provided.

Two earlier studies performed RNAseq analysis of Yap/Taz dKO SNs and provided information on transcripts regulated by YAP/TAZ (Deng et al., 2017; Poitelon et al., 2016). Given that Tead1 cKO basically phenocopies Yap/Taz dKO, showing less severe defects in developmental myelination, RNAseq analysis of Tead1 cKO will likely show overlapping patterns of dysregulated genes but unlikely to provide new mechanistic insights into how Yap/Taz-Tead1 regulate developmental myelination and remyelination. We will be more interested in using single cell RNAseq to elucidate how TEAD1 and other TEADs regulate non-myelinating Schwann cells or Remak bundles independently of YAP/TAZ. We will pursue it in our future studies of TEAD1-4 signaling in Remak bundle formation and homeostasis.

5. The effect of an induced Tead1 inactivation in SC is tested using a tamoxifen-inducible PLP specific Cre recombinase (Figure 7). However, data on the induced inactivation of Tead1 is missing. To validate this Tead1 iKO model, the authors must provide data (e.g., WB). Also, what is the percentage of tamoxifen-induced Tead1 iKO SC? The authors provide clear evidence for an effect on Krox20 in Yap1/Taz iDKO SC after 2 weeks of tamoxifen treatment. In Tead1 iKO SC, however, there is no effect after 6 weeks of tamoxifen treatment. How is this after 2 weeks? Could this be seen in Western blot analyses (c.f. to Tead1 cKO)? Also, how is the response to Oct6+ SC after 2 weeks? How is the response in Western blot analyses on targets of the cholesterol pathway (-> 2 weeks vs. 6 weeks)?

We previously provided the requested data on induced inactivation of Tead1 in Figure 1, because Tead1 iKO was used at first for Figure 2F (now removed). We now present the data as Figure 6-S1, which is placed next to Figure 7, so that readers will not miss the information. As the data indicate, our 2 weeks of tamoxifen administration protocol was highly efficient and removed TEAD1 from >95% Schwann cells in Tead1 iKO as shown by IHC. We also provide new WB data (Figure 6-S1A). Importantly, the residual TEAD1 that WB shows in Tead1 iKO is likely associated with non-Schwann cells, as indicated by IHC analysis.

We do not understand the rationale for examining the expression of Krox20 and Oct6 at 2 weeks in Tead1 iKO. Perhaps the reviewer considers it possible that Krox20 is rapidly lost in Tead1 iKO at 2 weeks causing demyelination, as in Yap/Taz iDKO (Grove et al., 2017), and then becomes restored at 6 weeks. As we stated, Tead1 iKO mice appeared normal 2 weeks after tamoxifen administration, unlike Yap/Taz iDKO, which displayed persistent shivering, ataxia, hunched posture, weight loss and rapid shallow breathing (Grove et al., 2017). Nonetheless, we examined Krox20, Oct6 and demyelination in Tead1 iKO at 2 weeks and found no differences from the results at 6 weeks, as expected.

6. Statistics section: Were all data tested for normal distribution (-> Shapiro Wilk test) before applying the indicated tests?

Yes, normal distribution of all the data was tested with Shapiro-Wilk normality test.

Reviewer #3 (Recommendations for the authors):The data are well-presented and solid. However, I have three suggestions that the authors might wish to address.1) I found the presentation of the inducible TeaD1 KO in Figure 1 very confusing. These mutants are analysed and presented in Figures 7 and 8. Why not introduce them there? The point of Figure 1 is mainly to demonstrate the effective deletion of TeaD1 from the Schwann cell lineage and its effect on the other TEAD proteins expressed in wild type and in the absence of TEAD1 (Figure1A and 1D).

As suggested, we now introduce Tead1 iKO as a Supplemental Figure (new Figure 6—figure supplement 1). We previously introduced both Tead1 iKO and Tead1 cKO in Figure 1, because Tead1 iKO was used at first for the IP experiment in Figure 2F, which we have now removed from the manuscript.

2) The timelines in Figure 7 and 8 should include the age at which TAM injection is started to avoid confusion on the age of the mice.

TAM injection was started when TEAD1 iKO mice were 8 weeks old. As suggested, we now include this information in the timelines of Figure 7 (now Figure 6) and Figure 8.

3) The authors should provide a much more detailed description of their YAP/TAZ IP experiments. In Figure 2F they present Western blots of IP of YAP/TAZ that appear close to 100% efficient, which is kind of incredible but possible of course. But what is really puzzling is the complete absence of a rabbit IgG heavy chain signal on their blots. They should explain the absence of this signal.

As discussed earlier, we have now removed Figure 2F and the IP experiment from the revised manuscript. Signal for rabbit IgG heavy chain is almost completely absent because we used a 2^nd^ antibody that was highly specific for IgG light chain.

Berti, C., Bartesaghi, L., Ghidinelli, M., Zambroni, D., Figlia, G., Chen, Z. L.,... Feltri, M. L. (2011). Non-redundant function of dystroglycan and beta1 integrins in radial sorting of axons. *Development, 138*(18), 4025-4037. doi:10.1242/dev.065490

Blanchard, A. D., Sinanan, A., Parmantier, E., Zwart, R., Broos, L., Meijer, D.,... Mirsky, R. (1996). Oct-6 (SCIP/Tst-1) is expressed in Schwann cell precursors, embryonic Schwann cells, and postnatal myelinating Schwann cells: comparison with Oct-1, Krox-20, and Pax-3. *J Neurosci Res, 46*(5), 630-640. doi:10.1002/(sici)1097-4547(19961201)46:5<630::aid-jnr11>3.0.co;2-0

Deng, Y., Wu, L. M. N., Bai, S., Zhao, C., Wang, H., Wang, J.,... Lu, Q. R. (2017). A reciprocal regulatory loop between TAZ/YAP and G-protein Galphas regulates Schwann cell proliferation and myelination. *Nat Commun, 8*, 15161. doi:10.1038/ncomms15161

Gerber, D., Pereira, J. A., Gerber, J., Tan, G., Dimitrieva, S., Yángüez, E., & Suter, U. (2021). Transcriptional profiling of mouse peripheral nerves to the single-cell level to build a sciatic nerve ATlas (SNAT). *ELife, 10*. doi:10.7554/*eLife*.58591

Grove, M., Kim, H., Santerre, M., Krupka, A. J., Han, S. B., Zhai, J.,... Son, Y. J. (2017). YAP/TAZ initiate and maintain Schwann cell myelination. *ELife, 6*. doi:10.7554/*eLife*.20982

Grove, M., Lee, H., Zhao, H., & Son, Y. J. (2020). Axon-dependent expression of YAP/TAZ mediates Schwann cell remyelination but not proliferation after nerve injury. *ELife, 9*. doi:10.7554/*eLife*.50138

He, X., Zhang, L., Queme, L. F., Liu, X., Lu, A., Waclaw, R. R.,... Lu, Q. R. (2018). A histone deacetylase 3-dependent pathway delimits peripheral myelin growth and functional regeneration. *Nat Med, 24*(3), 338-351. doi:10.1038/nm.4483

Monuki, E. S., Kuhn, R., & Lemke, G. (1993). Repression of the myelin P0 gene by the POU transcription factor SCIP. *Mech Dev, 42*(1-2), 15-32. doi:10.1016/0925-4773(93)90095-f

Pellegatta, M., De Arcangelis, A., D'Urso, A., Nodari, A., Zambroni, D., Ghidinelli, M.,... Feltri, M. L. (2013). alpha6beta1 and alpha7beta1 integrins are required in Schwann cells to sort axons. *J Neurosci, 33*(46), 17995-18007. doi:10.1523/JNEUROSCI.3179-13.2013

Poitelon, Y., Lopez-Anido, C., Catignas, K., Berti, C., Palmisano, M., Williamson, C.,... Feltri, M. L. (2016). YAP and TAZ control peripheral myelination and the expression of laminin receptors in Schwann cells. *Nat Neurosci, 19*(7), 879-887. doi:10.1038/nn.4316

Ryu, E. J., Wang, J. Y., Le, N., Baloh, R. H., Gustin, J. A., Schmidt, R. E., & Milbrandt, J. (2007). Misexpression of Pou3f1 results in peripheral nerve hypomyelination and axonal loss. *J Neurosci, 27*(43), 11552-11559. doi:10.1523/JNEUROSCI.5497-06.2007

Yu, W. M., Feltri, M. L., Wrabetz, L., Strickland, S., & Chen, Z. L. (2005). Schwann cell-specific ablation of laminin gamma1 causes apoptosis and prevents proliferation. *J Neurosci, 25*(18), 4463-4472. doi:10.1523/jneurosci.5032-04.2005